

# Global GOSAT, OCO-2 and OCO-3 Solar Induced Chlorophyll Fluorescence Datasets

Russell Doughty[1*], Thomas Kurosu[2], Nicholas Parazoo[2], Philipp Köhler[1], Yujie Wang[1], Ying Sun[3], Christian Frankenberg[1,2]

[1] Division of Geological and Planetary Sciences, California Institute of Technology, Pasadena, CA, 91125, USA
[2] Jet Propulsion Laboratory, Tropospheric Composition, Pasadena, CA, 91109, USA
[3] Soil and Crop Sciences Section, School of Integrative Plant Science, Cornell University, Ithaca, NY, 14853, USA

*Correspondence to*: Russell Doughty (rdoughty@caltech.edu)

**Abstract.** The retrieval of solar induced chlorophyll fluorescence (SIF) from space is a relatively new advance in Earth observation science, having only become feasible within the last decade. Interest in SIF data has grown exponentially, and the retrieval of SIF and the provision of SIF data products has become an important and formal component of spaceborne Earth observation missions. Here, we describe the global Level 2 SIF Lite data products for the Greenhouse Gases Observing Satellite (GOSAT), the Orbiting Carbon Observatory-2 (OCO-2), and OCO-3 platforms, which are provided for each platform in daily netCDF files. We also outline the methods used to retrieve SIF and estimate uncertainty, describe all the data fields, and provide users the background information necessary for the proper use and interpretation of the data, such as considerations of retrieval noise, sun-sensor geometry, the indirect relationship between SIF and photosynthesis, and differences among the three platforms and their respective data products. OCO-2 and OCO-3 have the highest spatial resolution spaceborne SIF retrievals to date, and the target and snapshot area mode observation modes of OCO-2 and OCO-3 are unique. These modes provide hundreds to thousands of SIF retrievals at biologically diverse global target sites during a single overpass, and provide an opportunity to better inform our understanding of canopy-scale vegetation SIF emission across biomes.

## 1 Introduction

Chlorophyll fluorescence is light that is emitted from chlorophyll after the absorption of photosynthetically active radiation (PAR), which covers the spectral range of roughly 400 to 700 nm and corresponds to the range of light visible to the human eye. The fluorescence emission  occurs in the range of ~650 to 800 nm during the light reaction of photosynthesis, where energy absorbed by leaf pigments is converted into the chemical energy that is needed by the dark reactions for fixing atmospheric carbon dioxide into sugars. The absorption of a photon by chlorophyll excites an electron, and the excitation energy has three main pathways: photochemistry, non-photochemical quenching or heat, and chlorophyll fluorescence. Most of



the excitation energy is used for photochemistry when vegetation is not stressed and light conditions are
not extreme, but at all times only a small fraction (~0.5-2%) is emitted as chlorophyll fluorescence (Porcar-
Castell et al., 2014; Maxwell and Johnson, 2000).

Chlorophyll fluorescence has been a research tool for studying photosynthesis for nearly 150 years (Müller,
1874), but only recently have spaceborne retrievals of solar induced chlorophyll fluorescence (SIF) been
realized (Guanter et al., 2007; Joiner et al., 2011; Frankenberg et al., 2011b). The number of spaceborne
platforms from which SIF can be retrieved continues to grow, and the SIF temporal record continues to
lengthen. Spaceborne SIF data has generated much excitement in a plethora of fields within the biological,
biogeochemical cycle, climate, and Earth system science communities. Chlorophyll fluorescence has long
been a key component of the plant physiological and ecophysiological research communities (Maxwell and
Johnson, 2000) and has traditionally been studied *in vivo* at the subcellular and leaf level, and *in situ* using
pulse amplitude-modulated (PAM) fluorometry (Schreiber et al., 1986).

Most recently, remote sensing techniques have enabled the canopy and ecosystem-level retrieval of SIF
from towers, aircraft, and satellites. The evolution in our ability to retrieve SIF infrequently at the leaf-level
to frequent  canopy-level retrievals across regional to global scales continues to greatly advance our
understanding of plant and ecosystem function and carbon cycling. However, there are fundamental
differences between in-situ PAM fluorometry and SIF. The former measures steady-state and light-
saturated fluorescence yields, which allow the derivation of photosynthetic yields (Genty et al., 1989) while
the latter only measures absolute SIF, following absorption of solar light by chlorophyll. The relationship
of SIF with photosynthetic yields is thus more complex (Porcar-Castell et al., 2014; Frankenberg et al.,
2014; Gu et al., 2019).

Here, we describe, compare, and discuss the Level 2 SIF Lite version 10 (v10) data produced from three
spaceborne platforms: the Greenhouse Gases Observing Satellite (GOSAT), the Orbiting Carbon
Observatory-2 (OCO-2), and OCO-3 (OCO-2 Science Team et al., 2020; OCO-3 Science Team et al.,
2020). Our data description is an update and synthesis of information that has been dispersed among several
user guides, publications, and supplementary materials related to these three platforms. Our presentation
and comparison of the SIF data from the three platforms and our discussions on SIF are intended to help
the user community find creative ways to apply the data and prevent misinterpretation.

Level 2 data is ungridded (vector) data that contains geophysical variables that are of interest and use to the
broader scientific community and is at same resolution of the Level 0 and Level 1 data, which are data



obtained as-is from the sensor (Level 0) to which ancillary information is appended (Level 1), such as
radiometric and geometric calibration coefficients and georeferencing parameters. Level 3 products refer
to gridded (raster) data, which can be found at https://climatesciences.jpl.nasa.gov/sif/download-data/level-

68  3/.


The annual and monthly spatial distribution of the Level 2 data for the globe and the continental United
States are presented in Figures 1 and 2 for visualization. These data are produced by the OCO-2 and OCO-
3 projects at the Jet Propulsion Laboratory (Frankenberg et al., 2014), quality controlled by NASA's
Making Earth System Data Records for Use in Research Environments (MEaSUREs) SIF team, and are
publicly available on the NASA Goddard Earth Sciences Data and Information Services Center (GES-
DISC) website (https://disc.gsfc.nasa.gov/). Recent efforts by the OCO and MEaSUREs team have focused
on harmonizing the processing pipeline, attributes, and file structures of the GOSAT and OCO SIF products
(Parazoo et al., 2019). Here, we present a first analysis of these harmonized products and demonstrate for
the user community their key commonalities and differences.

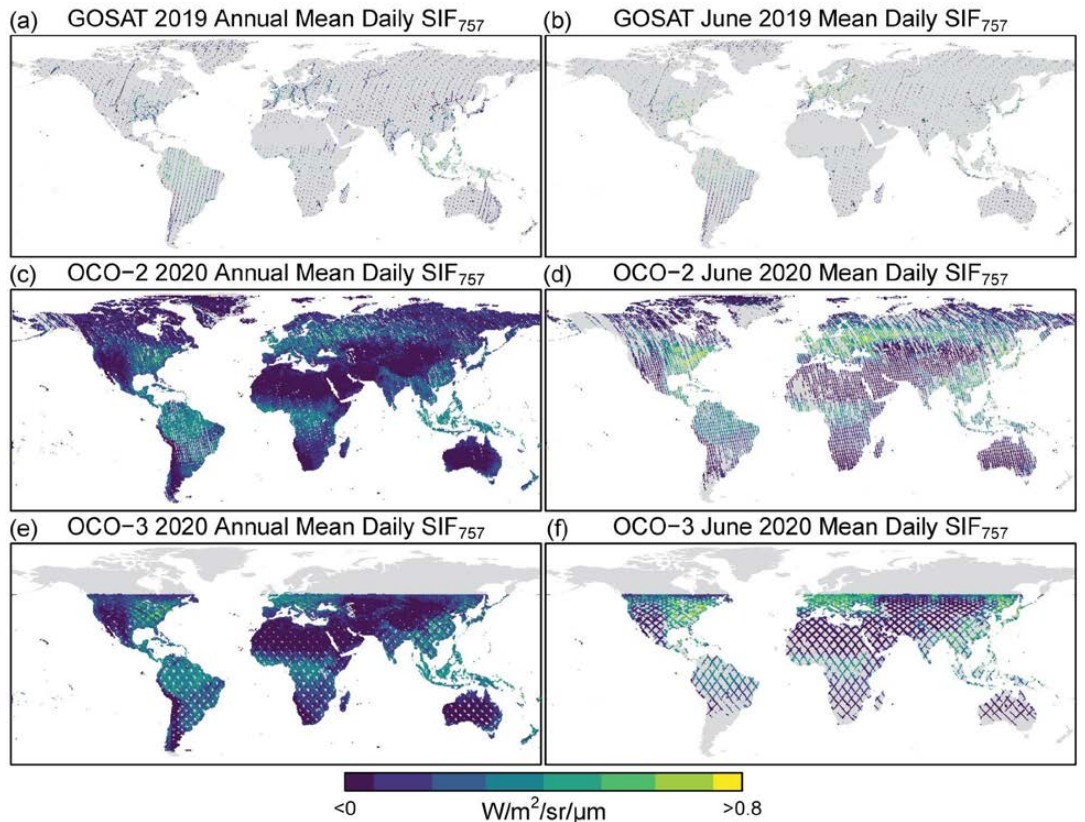




**Figure 1. Annual 2020 and June 2020 Mean Daily SIF$_{757}$ for GOSAT, OCO-2, and OCO-3.** The annual
and monthly coverage of GOSAT, OCO-2, and OCO-3 is presented here as mean daily SIF at 757 nm
(SIF$_{757}$) at a gridded resolution of 0.5° for visualization. Included are soundings from all measurement
modes flagged as *best* and *good* quality and *clear* of clouds. At nadir, the diameter of the GOSAT soundings
is ~10 km, and the widths of the OCO-2 and OCO-3 swaths are about 10 km and 13 km, respectively. Thus,
the data gaps shown here are larger than depicted and are not to scale.

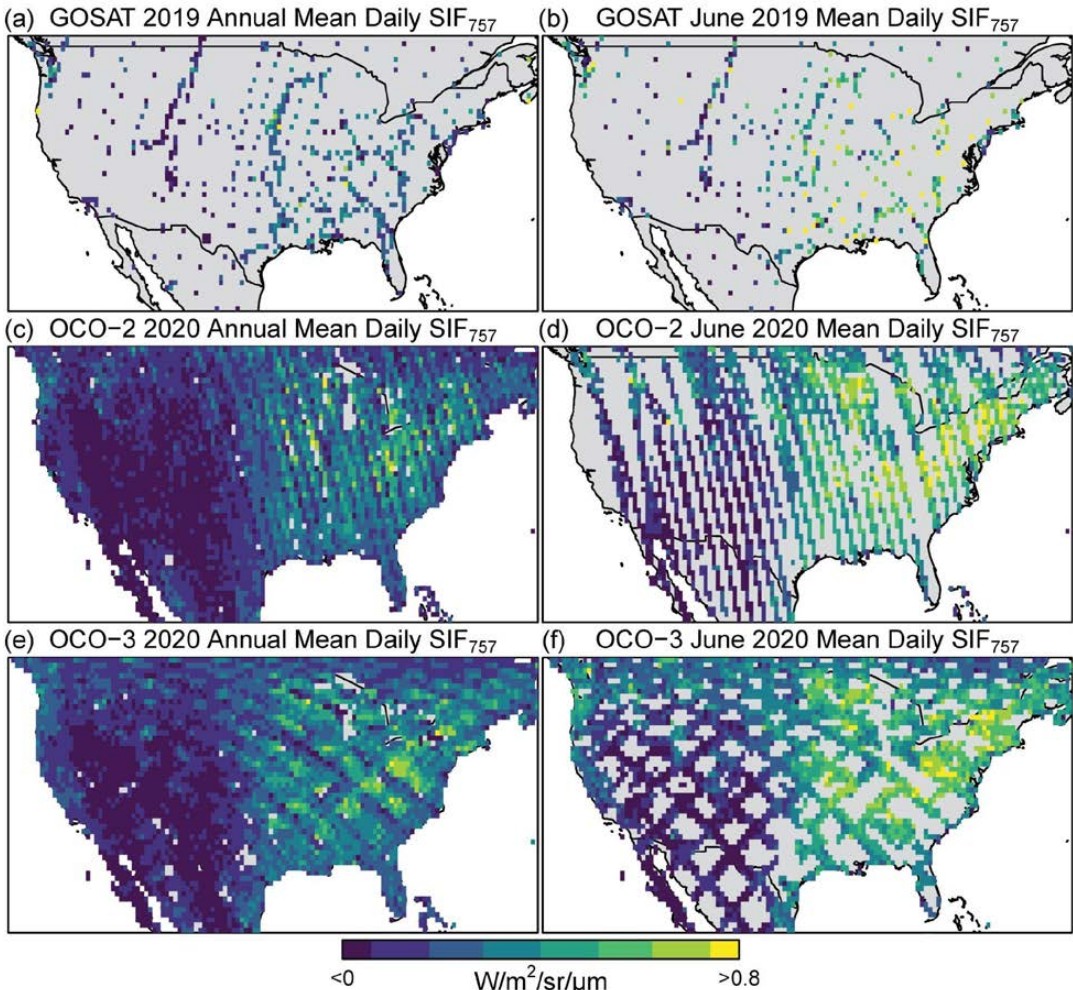


**Figure 2. Annual 2020 and June 2020 Mean Daily SIF$_{757}$ for GOSAT, OCO-2, and OCO-3 for**
**CONUS.** These panels are zoom-ins of the contiguous United States from Figure 1.



## 2 Satellite platforms

The retrieval of SIF requires high spectral resolution and a high signal to noise ratio (SNR) as solar Fraunhofer lines are very narrow and because SIF is a relatively weak signal (Frankenberg et al., 2011b). Coincidentally, the spaceborne spectrometers that have been used for retrieving Earth's atmospheric carbon dioxide and methane concentrations include spectral channels covering Fraunhofer lines in the vicinity of the oxygen A-band where atmospheric mass is retrieved with high spectral resolution (< 0.2 nm), enabling SIF retrievals with single measurement precision around ~0.5 $W/m^2/s/\mu m$. Thus, the retrieval of SIF from space has been pioneered by the atmospheric science community (Guanter et al., 2007; Joiner et al., 2011; Frankenberg et al., 2011b), and spaceborne SIF retrievals and data products have historically been a by-product of missions that have aimed to monitor Earth's atmospheric trace gases.

### 2.1 GOSAT

GOSAT (aka Ibuki) was developed by the Japan Aerospace Exploration Agency (JAXA) and launched in January 2009. In fact, the first global satellite SIF observations came from GOSAT (Joiner et al., 2011; Frankenberg et al., 2011b) (Joiner et al., 2011; Frankenberg et al., 2011b). Onboard the satellite is the greenhouse gas observation sensor (TANSO-FTS), which has a spectral resolution of 0.2 $cm^{-1}$ and an oxygen A-band SNR > 300. It has a sun synchronous, descending orbit with an overpass time of 13:00 ± 15 minutes at the equator, a 3-day repeat cycle, and a circular footprint of ~82 $km^2$ per sounding (~10 km diameter) (Kuze et al., 2009).

### 2.2 OCO-2 and OCO-3

OCO-2 is a NASA satellite that was launched in July 2014, and OCO-3 is a duplicate of the OCO-2 grating spectrometer attached to the Japanese Experimental Module Exposed Facility (JEM-EF) on the International Space Station (ISS) in May 2019 (Eldering et al., 2019). Each platform houses a 3-channel grating spectrometer with a spectral resolving power of $\lambda/\Delta\lambda$ >17,000 and a signal-to-noise ratio of >400 (Crisp et al., 2017; Eldering et al., 2019). They have three bands: an oxygen-A band at 0.765 μm and carbon dioxide bands at 1.61 μm and 2.06 μm. The swath widths are ~10 km with eight measurements across-track. The spatial resolution at nadir is slightly different for OCO-2 and OCO-3, about 1.3 km by 2.25 km and 1.6 km by 2.2 km, respectively.

OCO-2 has a 98.8-minute orbit with a 1:36 PM nodal crossing time and a 16-day ground-track repeat cycle (Crisp et al., 2017). The ISS has a precessing low-inclination orbit that allows OCO-3 to view Earth at absolute latitudes less than ~52°. The ISS orbits the Earth ~15.5 times a day and data acquisition is sometimes halted during ISS maintenance and docking, thus overpass times, revisit periods, and data



availability are relatively irregular. Validation of the OCO-2 SIF retrievals was conducted by Sun et al.
(2017) by comparing OCO-2 SIF to coordinated airborne measurements using the Chlorophyll
Fluorescence Imaging Spectrometer (Frankenberg et al., 2018).
**2.3 Observation Modes**
GOSAT observation modes are described as Observation Mode 1 Sunshine (OB1D), Observation Mode 2
Sunshine (OB2D), and Specific Observation Mode Sunshine (SPOD). OB1D is the routine observation
mode, whereas OB2D is a non-routine mode in which the thermal-infrared observation and pointing
mechanism is stopped during low power supply. Over land, SPOD is a target observation mode designed
to observe specific sites. The TANSO-FTS sensor has a setting for low, medium, and high gain. The
medium gain data is recommended for scenes that are bright, such as deserts. Since the data used for SIF
retrievals are filtered to exclude bright scenes due to deserts, ice, snow, and cloud cover, the high gain data
is used for SIF retrievals.

Nadir, glint, target, and transition observation modes are common to each OCO platform. The OCO-2 target
mode provides repeated spatial sampling of a given target, such as an emission source or tower site. The
OCO-3 target mode is a sequence of adjacent and partially overlapping swaths that allow for increased
spatial sampling. The target modes for both platforms provide over $10^3$ soundings. OCO-3 has an additional
observation mode using its pointing mirror assembly (PMA), which allows for snapshot area mapping
(SAM) of targets of interest. SAMs are a series of scans of a target that are nearly adjacent and can cover
an area of ~80 km by 80 km in about 2 minutes. The SAMs and their target locations, which include
volcanoes, various vegetation land cover types, and point sources of fossil fuel emissions, can be viewed at
https://ocov3.jpl.nasa.gov/sams/index.php. Target and SAM mode scans are prioritized and scheduled days
in advance of an overpass of the ISS over the target (Taylor et al., 2020).

The target and SAM observation modes offer unique, spatially resolved acquisition of a target during a
single overpass at different sun-sensor geometries as solar illumination is relatively fixed during overpasses
and soundings are acquired over a range of viewing angles as the sensors pass over their targets. For SIF
applications, these measurements can be averaged to obtain SIF estimates with a reduced standard error or
binned by sun-sensor geometries to investigate the effect of observation geometry of the retrieved SIF
values, as we demonstrate below.



## 3 Data description

### 3.1 SIF Lite file structure and content

The ungridded Level 2 SIF Lite data are provided in netCDF-4 format and contain information for each sounding from which a SIF retrieval was made. For each of the three satellite platforms, there is one file for each day in which there is at least one sounding and each file contains information for all soundings acquired on that day, including all measurement modes (glint, nadir, target). The SIF Lite files can be read by, but are not limited to, MATLAB, Python, R, and Julia using their respective netCDF4 or HDF5 libraries. The filename convention is, using the filename "oco2_LtSIF_200201_20210129t071949z.nc4" as an example, platform (oco2), data product (LtSIF), date (YYMMDD), and file creation date (YYYYMMDD) and time (tHHMMSS). The SIF Lite netCDF global attributes, dimensions, variables, and variable groups are described below and listed in Table1.

### 3.1.1 Global attributes and dimensions

The global attributes provide file-level metadata information, the most important of which for data users are the citation, contact information, and the time range of the data in the file. The times listed in the global attributes can be used in instances where the file names may have been changed. A netCDF dimension is an integer that specifies the shape of the multi-dimensional variables, and these are also described in Table 1. For the OCO-2 and OCO-3 data, there are dimensions for the footprint vertices (vertex_dim) and across-track footprint (footprint_dim), which are not applicable for GOSAT. The polarization dimension (polarization_dim) is used for GOSAT's P and S polarizations. The only variable dimension is the *sounding_dim*, which is the number of soundings in the file.

### 3.1.2 Variables

The primary variables of interest in the SIF Lite files are the *SIF*, *Daily_SIF*, and *SIF_Uncertainty* variables, which are available for SIF retrievals at 757 nm and 771 nm and estimated SIF at 740 nm. The variables for GOSAT differ from those of OCO-2 and OCO-3 in that GOSAT has two polarizations, P and S, and thus retrieval-related variables are provided as a 2-dimensional (2D) array. It is important to note that although the SIF values have traditionally been loosely labelled as being retrieved at 757 nm and 771 nm, the retrieval fit windows used to produce the SIF Lite data is centered at 758.7 and 770.1 for OCO-2 and OCO-3, and at 758 and 771 for GOSAT. However, we retain the 757 and 771 nomenclature to remain consistent with previous publications and to avoid confusion.

### 3.1.3 Variable groups

Most of the variables have been grouped, as listed in Table 1. The ungrouped, root-level variables are those that are most used and some of these variables are duplicated in the *Geolocation* and *Science* groups. The *Cloud* group contains cloud and surface albedo variables from the L2ABP product, which are used in the assignment of the quality flag. The *Geolocation* group contains variables related to the geolocation of the sounding footprint, sun-sensor geometry, altitude, and acquisition time. GOSAT sounding footprints are circular and have a radius of 5 km, in contrast to the OCO-2 and OCO-3 soundings, which are rhomboidal and are described with coordinates for each of their four vertices. Thus, the GOSAT SIF Lite files do not contain the footprint latitude and longitude vertices, whereas the OCO-2/3 SIF Lite files do.

The *Metadata* group houses variables with sounding-level metadata information, including build version of the data, unique orbit and sounding identifiers, and measurement mode.

The *Meteo* group contains meteorological forecast variables, which were obtained from the GEOS-5 FP-IT 3h forecast (Lucchesi, 2015) and are provided as-is without validation. The *Offset* group is a collection of variables of the bias/offset adjustments and statistics. These include mean, median, and standard deviations of the adjusted and unadjusted SIF values separated by cross-track footprint. These data are reported on a grid of signal level bins with a range of 3.0-229.0 $W/m^2/s/\mu m$ and follows the SIF bias correction scheme outlined by (Frankenberg et al., 2011b).

### 3.2 Quality flag criterion and rationale

The Quality_Flag variable indicates the quality of the data for each sounding as being *best* (0), *good* (1), or *failed* (2). We recommend using a combination of *best* and *good* for scientific analysis. The criterion for the *best* and *good* quality flags are listed in Table 2, and soundings that do not meet either set of criteria are flagged as *failed*. The rationale for the criterion is as follows: reduced chi-square ($\chi^2$) thresholds exclude fits that do not well represent the spectrum; continuum level radiance excludes scenes with brightness that is too high or low; solar zenith angle ($\theta$) excludes retrievals with extreme solar zenith angles, which are more likely affected by rotational Raman scattering; and the $O_2$ and $CO_2$ thresholds exclude most cloudy scenes.

## 4 Methods

### 4.1 SIF retrieval

The SIF values provided in the SIF Lite files are based on spectral fits covering Fraunhofer lines, as SIF reduces the fractional depth of the Fraunhofer lines (Plascyk, 1975). The SIF retrieval methodologies are fully explained by Frankenberg et al. (2011b, a) and SIF is retrieved for GOSAT and the OCO platforms at 757 nm and 771 nm. We estimated SIF at 740 nm for each sounding using both retrieval windows as described in more detail below. The main retrieval quantity in the retrieval state vector is the fractional contribution of SIF to the continuum level radiance, or relative fluorescence (SIF_Relative_757nm and SIF_Relative_771nm). The absolute SIF values (SIF_757nm and SIF_771nm) are generated during post-processing in $W/m^2/s/\mu m$.

### 4.2 SIF 740 nm and intersensor comparisons

The spectral window in which SIF retrievals are made depends on the wavelength bands of the platform. Assuming the spectral shape of SIF is known and invariant, one can convert SIF to a standard reference wavelength. Here, we use 740 nm as a reference as it corresponds to the 2nd SIF peak and is not as strongly affected by chlorophyll re-absorption as red SIF, thus showing a relatively stable shape at wavelengths above 740 nm. The differences in the retrieval windows complicate the comparison of SIF retrievals from different sensors, thus it is useful to provide SIF at a well-defined reference wavelength.

Although the range of the wavelengths used to retrieve SIF from the various sensors is small (740-771 nm), absolute fluorescence can vary greatly depending on the spectral window used to retrieve SIF (Joiner et al., 2013; Köhler et al., 2018; Sun et al., 2018). However, reference far-red SIF emission spectra at the leaf level indicates that far-red fluorescence spectral shapes are consistent across species (Magney et al., 2019). Thus, we provide an estimate of absolute $SIF_{740}$ (SIF_740nm) in the GOSAT and OCO-2/3 SIF Lite files derived from the empirical relationship between SIF at 740 nm and SIF at 758.7 nm and 770.1 nm (denoted as 757 nm and 771 nm; Eq. 1). The rationale for including $SIF_{740}$ in the SIF Lite files is to allow for more consistent and robust comparisons of SIF and SIF-based analyses across sensors (Parazoo et al., 2019), and to reduce the retrieval error by a factor of $\sqrt{2}$ (Sun et al., 2018).

$$SIF_{740} = 0.5 \cdot (1.5 \cdot SIF_{757} + 2.25 \cdot SIF_{771}) \tag{1}$$

We noted that although the empirical ratio of $SIF_{757}$ and $SIF_{771}$ is 1.80 based on leaf level measurements conducted by Magney et al. (2019), we observed a median ratio of 1.45 from OCO-2 over vegetated areas





for 2015-2019 (Figure S1). The reason for this difference has not yet been discerned and requires further
analysis, but the small potential bias introduced by the use of the empirical ratio does not infringe on the
utility of the $SIF_{740}$ data.
**4.3 SIF retrieval uncertainty**
The determination of single sounding retrieval uncertainty is covered in great detail by Sun et al. (2018)
and Frankenberg et al. (2014), and is provided in the SIF Lite files as SIF_Uncertainty_740nm,
SIF_Uncertainty_757nm, and SIF_Uncertainty_771nm. Briefly, these values are the 1-sigma ($\sigma$) estimated
single sounding measurement precision and represent the random component of the retrieval errors. It is
derived through standard least-square fitting by evaluating the error covariance matrix:

$$S_e = (K^T S_0 K)^{-1} \tag{2}$$



where $K$ is the Jacobian matrix of the least-squares fit, and $S_0$ is the measurement error covariance matrix,
which characterizes the instrument noise per detector pixel.

For OCO-2/3, the uncertainty for $SIF_{757}$ usually ranges between 0.3 and 0.5 W/m²/s/μm, or ~15-50% of the
absolute SIF value. Uncertainties for $SIF_{771}$ are slightly higher due to less fluorescence and a relatively less
reduction in the fractional depth of the radiance at 771 nm. Uncertainty for $SIF_{740}$ is calculated from $SIF_{757}$
and $SIF_{771}$:

$$SIF_{Uncertainty_{740}} = 0.5 \cdot \sqrt{\left( \left( 1.5 \cdot SIF_{Uncertainty_{757}} \right)^2 + \left( 2.25 \cdot SIF_{Uncertainty_{771}} \right)^2 \right)} \tag{3}$$


**4.4 Bias/offset correction**
Biases in retrieved SIF can occur due to uncertainties in the exact instrument line-shape per footprint or
slight uncertainties in detector linearity. To correct for biases, we use reference targets that are non-
fluorescent surfaces barren of vegetation, similar to the method described by Frankenberg (2011b). In short,
the background signal over reference targets is subtracted from all relative SIF values. We calculate the
background signal for each day as mean SIF over all barren surfaces within a 31-day window centred on
the current day for GOSAT and a 3-day window for OCO-2/3. Here, we identify barren surfaces using a
combination of the MODIS MCD12Q1 land cover data product (Friedl and Sulla-Menashe, 2019) and the
Vegetation Photosynthesis Model (VPM) (Xiao et al., 2004; Zhang et al., 2017) from the year 2018. The
native spatial resolution of these data sets is 500 m, but we aggregated the data to a global 0.20-degree grid
so that the barren surface reference targets had a coarser resolution than the soundings. We classified barren
surfaces as those grid cells which were 100% barren and/or snow and ice by MCD12Q1 and had zero (0)
annual gross primary production as estimated by VPM. We also excluded coastal grid cells that overlapped
with water using a global coastline shapefile and a buffer.
**4.5 Daily average SIF and the daily correction factor**
We provide an estimate of daily average SIF (Daily_SIF), which is instantaneous SIF scaled entirely upon
the geometry of incoming solar radiation over a day. Instantaneous SIF is the absolute value of SIF for any
given sounding and is a strong function of the illumination of the canopy at that instant in time. The
differences in the illumination geometry of soundings at different overpass times and latitudes complicate
direct comparisons of SIF at different points of Earth's surface and comparisons of SIF to other data that
are more temporally coarse, such as daily estimates of GPP.

Downwelling solar radiation scales linearly with $cos(\theta)$ under clear sky conditions when ignoring Rayleigh
scattering and gas absorption. As described by Frankenberg et al. (2011b) and Köhler et al. (2018), a first
order approximation of daily average SIF ($SIF_{Daily}$) can be written as:
$$SIF_{Daily} = SIF_{t0} \cdot \frac{1}{\cos(\theta(t_0))} \cdot \int_{t=t_0-12h}^{t=t_0+12h} cos(\theta(t)) \cdot H\left(cos(\theta(t))\right) dt \qquad (4)$$
ere $SIF_{t0}$ is absolute instantaneous SIF, $\theta(t_0)$ is the solar zenith angle $\theta$ at the time of measurement $t_0$ with
a heaviside function H to zero out negative values of $cos(\theta)$, and the integral is computed numerically
in 10-min time steps ($dt$). In terms of the SIF Lite file variable names, this equation can be written for SIF
at any wavelength as $Daily\_SIF = SIF \cdot daily\_correction\_factor$.
**5 Discussion**
**5.1 Scaling of SIF to GPP**
We should note that SIF is, to first order, only a proxy for the electron transfer rate in the light reaction of
photosystem II. However, SIF is oblivious to the light-independent reactions that fix $CO_2$. Nevertheless,
many studies have reported on the linearity of SIF and GPP at bi-weekly or monthly timescales and at
coarse spatial resolutions (Verma et al., 2017; Doughty et al., 2019; Yang et al., 2015). The seasonality of
SIF and GPP tend to match well at such coarse temporal resolutions because both SIF and GPP are being
driven by changes in canopy structure, the amount chlorophyll in the canopy, and the amount of sunlight
(photosynthetically active radiation; PAR) being absorbed by canopy chlorophyll ($APAR_{chl}$) (Magney et



al., 2020; Doughty et al., 2021; Dechant et al., 2019). The SIF-GPP relationship can also become more linear at the canopy scale due to the contribution of total canopy SIF by sunlit, shaded, stressed, and non-stressed leaves (Magney et al., 2019). SIF and GPP have an indirect relationship through non-photochemical quenching and the electron transport rate (Porcar-Castell et al., 2014; Gu et al., 2019), which can sometimes simultaneously downregulate photosynthesis and SIF, as has been seen in evergreen needleleaf ecosystems, but not always (Magney et al., 2019).

At the leaf level, GPP saturates before SIF in response to APAR, such that we could see increased SIF without any response in GPP at high levels of APAR (Gu et al., 2019). Conversely, vegetation stress can cause a near or total cessation of GPP via stomatal closure with little or no change in SIF. This decoupling has been seen at the leaf scale during forced stomatal closure of deciduous tree species (Marrs et al., 2020) and a 1-month drought experiment with Eastern cottonwood (*Populus deltoides*) (Helm et al., 2020). However, these studies and others of deciduous vegetation and croplands have repeatedly found a better correlation between SIF and APAR than SIF and GPP (Yang et al., 2018; Miao et al., 2018). For SIF to be a reliable proxy of APAR, $SIF_{yield}$ (ratio of SIF to APAR) would need to remain constant. For a detailed inquiry into SIF and photosynthesis, see Porcar-Castell et al. (2014), and a review of SIF remote sensing applications and challenges from the leaf, tower, and satellite scale by Magney et al. (2020) and Mohammed et al. (2019).

### 5.2 Negative SIF values

Data users are likely to find negative SIF values, which are due to retrieval noise, but these values should generally not be discarded. The one-sigma uncertainty in retrieved SIF values (SIF_Uncertainty) can be substantial, but negative values are plausible in a retrieval sense although not in physical terms (actual SIF emission cannot be negative). Discarding negative values will introduce a high bias when averaging. Nevertheless, extremely negative values may indicate a problem with the retrieval. We recommend the following guidelines for filtering negative SIF values: accept if SIF + 2-σ uncertainty ≥ 0; questionable if SIF + 2-σ uncertainty < 0 and SIF + 3-σ uncertainty ≥ 0; and reject if SIF + 3-σ uncertainty < 0. These thresholds have not been incorporated into the Quality_Flag variable of the SIF Lite data.

### 5.3 Sun-sensor geometry

Users of SIF data from any source should be aware that sun-sensor geometry plays a role in the absolute values of SIF, in addition to vegetation canopy characteristics (Joiner et al., 2020; Köhler et al., 2018). Absolute SIF values increase rapidly when the phase angle approaches 0° (when the sun and sensor are aligned), but the effect of sun-sensor geometry has been shown to be small when the phase angle is greater

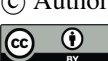

than 20° (Köhler et al., 2018; Doughty et al., 2019). Thus, retrieved SIF values from target or SAM mode
scans during a single overpass can vary greatly despite homogeneous vegetation cover due to changing sun-
sensor geometries during data acquisition. Figure 3 illustrates the phase angle and $SIF_{757}$ for a SAM
acquired over the Amazon rainforest, where the vegetation canopy is very homogenous. The figure also
illustrates how the phase angle changes during an OCO-3 SAM scan and that the sun-sensor geometries for
each individual swath are rather distinct from each other (Figure 3a). Mean SIF for each swath is also
distinctively different (Figure 3b), despite that the canopy was experiencing the same illumination geometry
and environmental conditions during the two minutes in which this SAM was acquired. The effect of sun-
sensor geometry is also illustrated in Figure 4, which shows the relationship between SIF for individual
OCO-2 soundings and phase angle for two target scans in the Amazon. A distinctive change in the absolute
values of retrieved SIF were observed due to sun-sensor geometry.

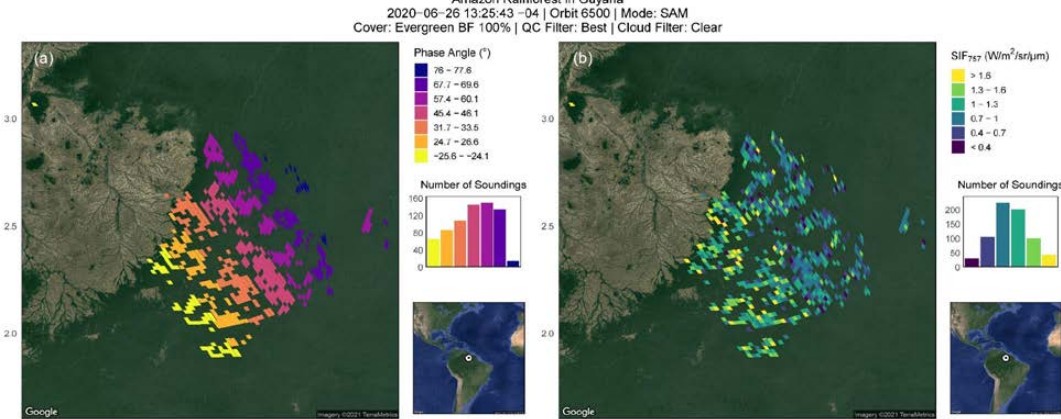

**Figure 3. Phase angle and $SIF_{757}$ for an OCO-3 SAM mode scan over the Amazon Rainforest in**
**Guyana.** OCO-3 SAMs are composed of several scans of a target whereby the eight-sounding wide swath
is offset adjacent to the previous scan. Each swath has a distinctive, small range of phase angles as seen in
(a). SIF has higher values at lower phase angles, which is apparent in (b) where the higher SIF values
occur for the soundings in the southwestern portion of the SAM where phase angles are lowest.



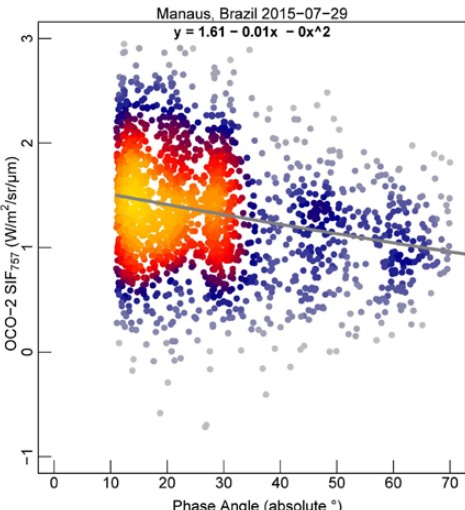


**Figure 4. Absolute phase angle and SIF$_{757}$ for an OCO-2 target mode scan over evergreen broadleaf**
**forest in Manaus, Brazil.** As this figure demonstrates, retrieved SIF values increase as the phase angle
approaches 0 degrees.

### 5.4 Averaging over space and time to reduce retrieval uncertainty

There are two main challenges to working with all spaceborne SIF data: 1) the inherently large uncertainties
for individual soundings due to retrieval noise, and 2) the effect of differences in sun-sensor geometry on
retrieved SIF values. Thus, we advise against using single soundings for analysis. However, averaging
soundings across space and time can reduce the retrieval noise by a factor of $1/\sqrt{n}$, with $n$ being the number
of soundings comprising the average (Frankenberg et al., 2014). For platforms with a wide swath, like the
TROPOspheric Monitoring Instrument (TROPOMI), the effect of sun-sensor geometry can be accounted
for by averaging soundings for a point of interest over the entire repeat cycle (16-days for TROPOMI) as
demonstrated by Doughty et al. (2019, 2021). In the case of OCO-2/3, as we demonstrate in Figure 3 and
in Braghiere et al. (2021), soundings can be grouped by phase angle and then averaged to reduce retrieval
uncertainty. Thus, retrieval uncertainty and sun-sensor geometry effects can be substantially minimized.
For GOSAT, we recommend averaging SIF retrieved from both the P and S polarizations, as demonstrated
in Figure 5.

Users should also keep in mind that when conducting analyses at large spatial scales, gridding the data prior
to analysis is largely unnecessary as the ungridded Level 2 data can be used directly (Doughty et al., 2019).
Doing so will allow the users to retain sounding-level information that may aid in the interpretation of the
results, which would otherwise be lost when merely gridding the SIF values.

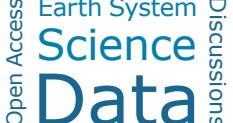

### 5.5 The use of SIF at 740, 757, and 771 nm

It is important to note that in areas where the SIF signal is weak, the use of $SIF_{757}$ at would be more appropriate as the SIF signal is stronger at this wavelength. In areas where vegetation is sparse or $SIF_{yield}$ is low due to vegetation responses to environmental conditions or canopy leaf physiology, $SIF_{771}$ could be within the noise range due to its relatively far distance from the far-red peak at 740 nm. In these cases, we advise the use of $SIF_{757}$. Since $SIF_{771}$ is used to compute $SIF_{740}$ in the SIF Lite files, diligence should likewise be used when using $SIF_{740}$ in analyses.

### 5.6 Comparison of GOSAT, OCO-2, and OCO-3

OCO-3 SIF has been shown to have a very high correlation ($r > 0.9$) with OCO-2 (Taylor et al., 2020). Here, we present the first comparisons between GOSAT and OCO-2 Level 2 data. Currently, there are not enough coincident soundings for GOSAT and OCO-3 to provide a robust analysis but given that OCO-2 and OCO-3 compare very well, we would expect a comparison between GOSAT and OCO-3 to mimic the findings from our GOSAT and OCO-2 comparison.

Although the data record for GOSAT and OCO-2 overlap six years, only a small percentage of soundings flagged as best quality and cloud free from GOSAT and OCO-2 overlap on the same day (Figure 5a). Despite this filter, the mean SIF values may differ widely on the same day due to differences in overpass time (and thus solar illumination angle and environmental conditions), viewing geometry, and the number of OCO-2 soundings comprising the mean. We progressively filtered the data as illustrated in Figure 5 to ensure the soundings were of a vegetated land surface, had similar sun-sensor geometries, environmental, and atmospheric conditions, and that the temperature was high enough for photosynthesis to occur.

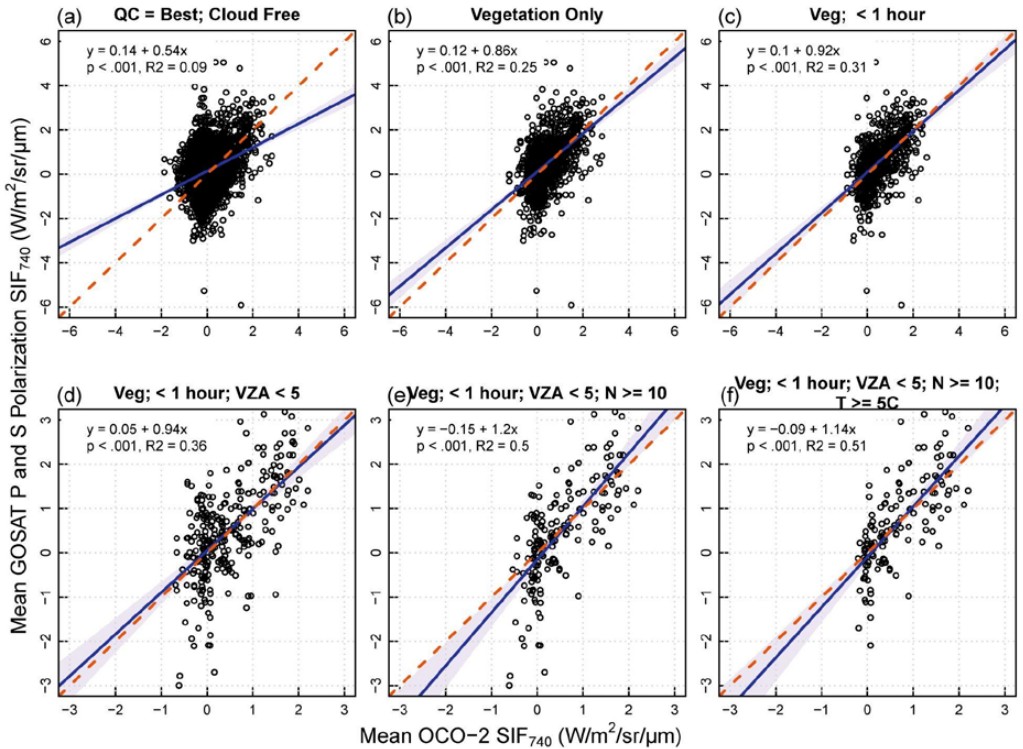

398

**Figure 5. Relationships of SIF$_{740}$ from OCO-2 and GOSAT using progressively conservative data**

**filters and Deming regression.** X-axis values are the mean of all OCO-2 soundings (~1.3 km by 2.25

km) that fall within the corresponding GOSAT sounding footprint (~10 km in diameter). Y-axis values

represent the mean of SIF retrieved from P and S polarizations for a single GOSAT sounding.  Six years

of data (2015-2020) were used to identify soundings that overlapped on the same day. (a) Soundings

flagged as best quality and cloud free. (b) Same as (a) but filtered as being over vegetation using the

IGBP flag in the OCO-2 SIF Lite file. (c) Same as (b) but filtered for data that was acquired from GOSAT

and OCO-2 within one hour of each other. (d) Same as (c) but with viewing zenith angles (VZA) < 5° for

both platforms. (e) Same as (d) but with number (N) of OCO-2 soundings within a GOSAT sounding

being ≥ 10. (f) Same as (e) but with skin temperature ≥ 5 °C.

We found that the correlation and slope improved with more conservative filtering of the data, and that the

comparison between GOSAT SIF and OCO-2 SIF were reasonable. However, it is important to note that

any comparison between GOSAT and OCO data will inevitably be affected by spatial sampling bias, as the

swath width for both OCO platforms is smaller than the diameter of the GOSAT footprints (Figure 6; left

footprints). Also, it could be the case that only a small portion of the GOSAT footprint is sampled by OCO





(Figure 6; right footprints). Our filter of ≥ 10 OCO-2 soundings within a GOSAT footprint aimed to reduce
this potential sampling bias in addition to reducing the uncertainty of the OCO-2 SIF retrievals. It must also
be remembered that in this comparison, we do not have the luxury to average several GOSAT soundings to
reduce the uncertainty as we did with OCO-2, so the uncertainties of the GOSAT SIF is much higher than
that for OCO-2.

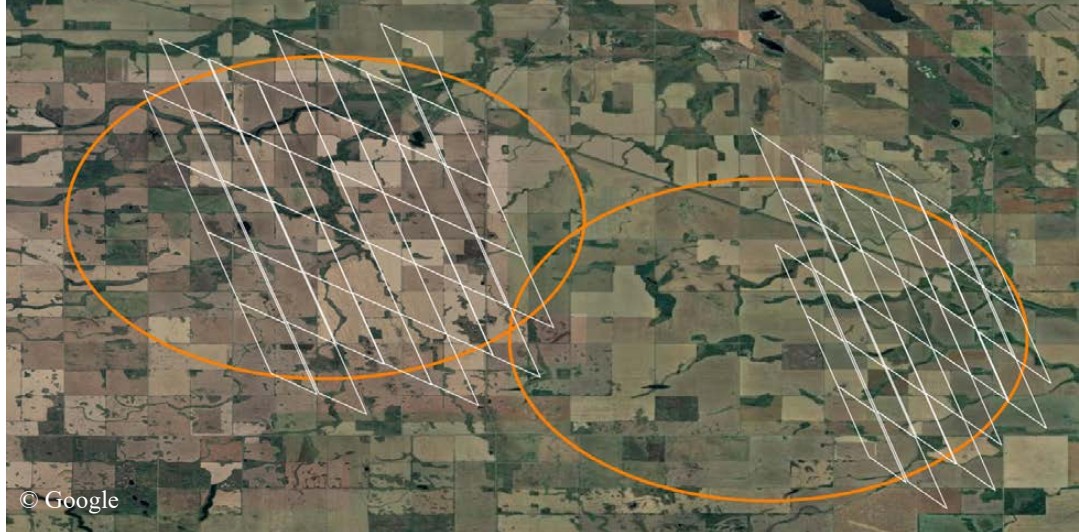


**Figure 6. Overlapping GOSAT and OCO-2 soundings near Quill Lakes, Saskatchewan, Canada.**
Orange circles are GOSAT sounding footprints (~10 km) and the white rhomboids are OCO-2 sounding
footprints (~1.3 km by 2.25 km) acquired on the same day as the GOSAT soundings in which they fall.
The GOSAT and OCO-2 soundings on the left were acquired in February 2019, and the soundings on the
right were acquired in July 2017. The base map is a Google Satellite image.

Upon a more detailed comparison of GOSAT and OCO-2 SIF at 740 nm, 757 nm, and 771 nm using the
strictest filter we applied in Figure 5f, we found $SIF_{740}$ from the two platforms to have higher correlations
than for $SIF_{757}$ and $SIF_{771}$ alone (Figure 7). We also noticed that GOSAT and OCO-2 soundings most
frequently overlap in the boreal winter, which corresponds to a period of little or no photosynthesis at mid
and high latitudes (Figures S2 and S3). Thus, the direct comparison of GOSAT and OCO-2 SIF is severely
restricted by the relatively infrequent overlap of the two platforms during the growing season.



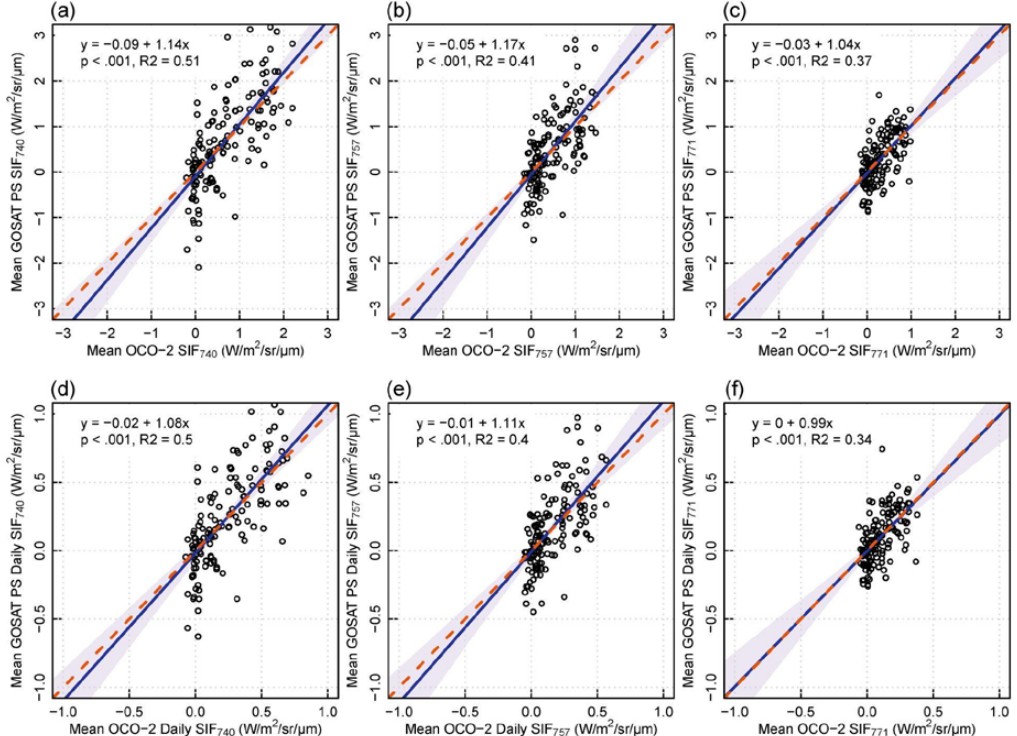

**Figure 7. Relationships between SIF$_{740}$, SIF$_{757}$, and SIF$_{771}$ from GOSAT and OCO-2 using Deming regression.** The soundings presented here were those presented in main text Figure 5f, which were data that had the most conservative filter: best quality and cloud free, vegetation, co-occurring within 1 hour, viewing zenith angle $< 5°$, number of OCO-2 soundings within a GOSAT footprint $\geq 10$, and skin temperature $\geq 5 °C$.

In addition to the sounding level comparisons, we found mean annual SIF$_{757}$ for GOSAT and OCO-2 to compare reasonably well at the global scale during the boreal summer (Figure 8). The relatively large differences in SIF illustrated at the gridcell level in Figure 8c are due to differences in the spatial and temporal sampling of the two platforms.

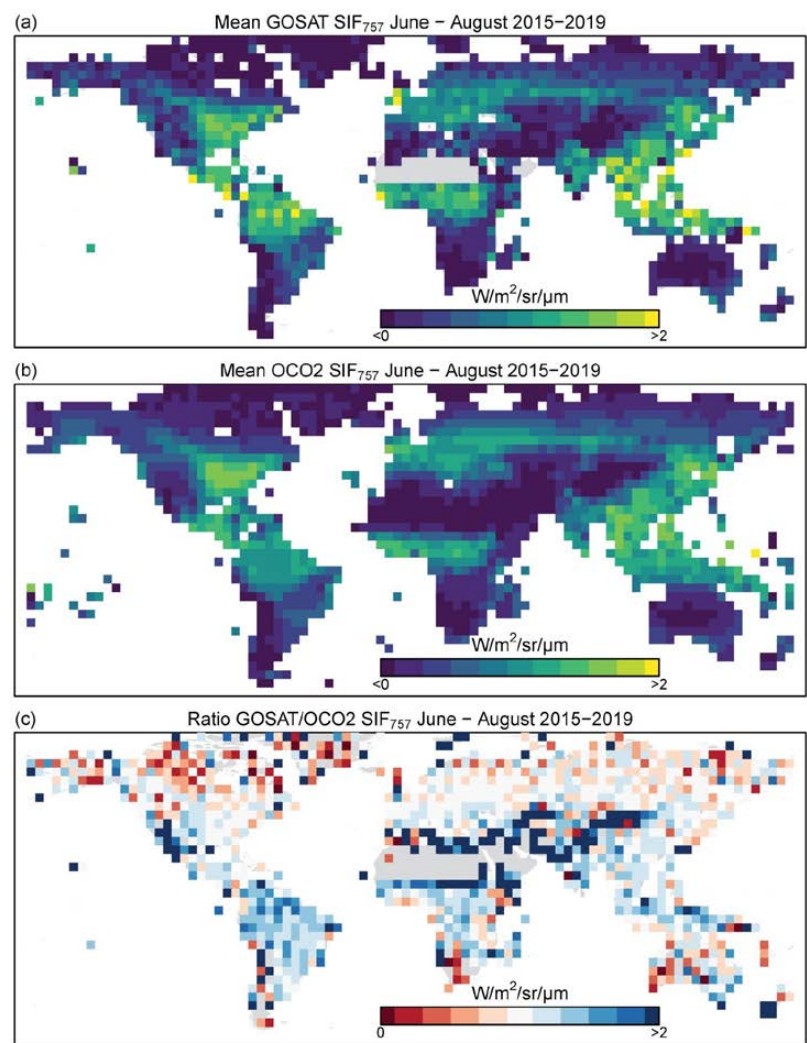

**Figure 8. Mean GOSAT to OCO-2 SIF$_{740}$ and their ratio at 4.0 degrees for June-August 2015-2019.**

**5.7 Collocating Soundings with their Targets**

Currently, the target and SAM soundings are not collated to the target to which they correspond, but variables will be added to future versions of the SIF Lite files that will allow for the collocation of target and SAM soundings with their intended target site. For OCO-3, some of the target sites are in close proximity to each other and thus a target site may fall within the scan of another target. For these sites, users may also want to check scans that were intended for target sites adjacent to their target of interest. The





OCO-3 targets, the dates of their scans, and scan maps are available at
https://ocov3.jpl.nasa.gov/sams/index.php. A list of target locations for OCO-2 and OCO-3 are available in
Table S1 and Table S2, respectively.
**6 Conclusions**
Users of remote sensing data are more accustomed to using Level 3 gridded data for analyses, but we
incentivize data users to also exploit the Level 2 data we have presented in the SIF Lite files. The OCO-2
and OCO-3 platforms provide the highest spatial resolution spaceborne SIF data, and the target and SAM
observation modes are unique to these platforms. The observation scheme for the OCO platforms allow for
time series to be constructed for the target locations, and the repeated target and SAM scans allow for the
investigation of the directionality and escape of SIF at varying sun-sensor geometries across many biomes
in different seasons.

We have demonstrated how users can break target and SAM observations into phase angles for analysis
and have described how the effect of sun-sensor geometry and retrieval noise can be mitigated through the
averaging of the data. The OCO platforms also provide a rich resource for the validation of radiative transfer
models, which is currently underutilized. Upcoming spaceborne platforms with frequent revisits and/or
high spatial resolution, such as the FLuorescence EXplorer (FLEX) by the European Space Agency and
NASA's Geostationary Carbon Cycle Observatory (GeoCarb), are expected to further our understanding of
changes in vegetation structure and function (Drusch et al., 2016; Polonsky et al., 2014; Moore et al., 2018).
**7 Data availability**
All SIF Lite files presented here can be found at NASA Goddard Earth Sciences (GES) Data and
Information Services Center (DISC) at https://disc.gsfc.nasa.gov/datasets/. OCO-2 can be accssed at
https://doi.org/10.5067/XO2LBBNPO010, and OCO-3 data can be accessed at
https://disc.gsfc.nasa.gov/datacollection/OCO3_L2_Lite_SIF_EarlyR.html. Links to other SIF data
products are listed at NASA Jet Propulsion Lab (JPL) website for SIF at
https://climatesciences.jpl.nasa.gov/sif/.
**8 Author contributions**
RD and CF conceived this manuscript. TK prepared and provided the data and RD performed the
analysis. RD prepared the manuscript with contributions from all co-authors.





**9 Competing interests**

The authors declare that they have no conflict of interest.

**10 Acknowledgements**

We thank Lan Dang for helping to process the GOSAT data and Annmarie Eldering for helping coordinate the publication of the SIF Lite files at the GES-DISC.

**11 Financial support**

This research was supported by NASA Making Earth System Data Records for Use in Research Environments (MEaSUREs) Program (NNN12AA01C) and the NASA OCO Science Team (80NSSC18K0895).

**Table 1. Level 2 GOSAT, OCO-2, and OCO-3 SIF Lite netCDF File Global Attributes, Dimensions, and Variables.** Units for SIF and continuum level radiance variables are W/m$^2$/sr/µm, geolocation variables are in decimal degrees, angles are in degrees, and the units for the meteorological variables are in the table below. For GOSAT, data is provided for both the P and S polarizations as a 2-dimensional array. * denotes the variable or dimension is only applicable to OCO-2 and OCO-3, and ** denotes that the dimension is only applicable to GOSAT. Note that there are different MeasurementMode and OrbitID descriptions for GOSAT, and that some root-level variables are duplicated in the Geolocation and Science group.

| Global Attributes | |
| --- | --- |
| date_time_coverage | UTC time string of the first and last observation |
| day_of_year_coverage | Same as date_time_coverage, but with day-of-year |
| InputCollectionLabel | Collection label of the L2 data products used to create the file |
| InputBuildID | Build ID of the L2 data products used to create the file |
| InputPointers | String with names of all input products and auxiliary data used to create the file |
| **Dimensions (length of dimension)** | |
| sounding_dim (variable) | Number of soundings in the file |
| footprint_dim (8) * | Number of OCO-2/3 across-track footprints |
| vertex_dim (4) * | Number of footprint corner coordinates |
| signalbin_dim (227) | Number of entries in the signal histogram arrays in the Offset group |
| statistics_dim (2) | Array dimension in the Mean and Median SIF values of the Offset group; adjusted and unadjusted values |
| polarization_dim (2) ** | Array dimension of the polarization for GOSAT; P and S polarization |



| Root Level Variables | |
| --- | --- |
| Daily_SIF_740nm | Daily Corrected Solar induced chlorophyll fluorescence at 740 nm: Daily_SIF_740nm = SIF_740 * /Science/daily_correction_factor |
| Daily_SIF_757nm | Daily Corrected Solar induced chlorophyll fluorescence at 757 nm: Daily_SIF_757nm = /Science/sif_757nm * /Science/daily_correction_factor |
| Daily_SIF_771nm | Daily Corrected Solar induced chlorophyll fluorescence at 771 nm: Daily_SIF_771nm = /Science/sif_771nm * /Science/daily_correction_factor |
| Delta_Time | Timestamp (seconds since 1 January 1990) |
| Latitude | Center latitude of the measurement |
| Latitude_Corners * | Corner latitude of the measurement |
| Longitude | Center longitude of the measurement |
| Longitude_Corners * | Corner longitude of the measurement |
| Quality_Flag | 0 = best (passes quality control + cloud fraction = 0.0); 1 = good (passes quality control); 2 = bad (failed quality control); -1 = not investigated |
| SAz | Azimuth angle between the solar direction as defined by the sounding location, and the sounding local north |
| SIF_740nm | Solar induced chlorophyll fluorescence at retrieved wavelength: SIF_740nm = 0.75 * (/Science/sif_757nm + 1.5*/Science/sif_771nm) |
| SIF_Uncertainty_740nm | Uncertainty computed from continuum level radiance at 740 nm: $SIF\_Uncertainty\_740 = 0.75 * ((/Science/sif\_757nm)^2 + (1.5*/Science/sif\_771nm)^2)^{(1/2)}$ |
| SZA | Solar zenith angle is the angle between the line of sight to the sun and the local vertical |
| VAz | Azimuth angle between line of sight and local north |
| VZA | Sensor zenith angle is the angle between the line of sight to the sensor and the local vertical |
| Variable/Group Name | Description |
| **Cloud Group Variables** | |
| cloud_flag_abp | Indicator of whether the sounding contained clouds: 0 - Classified clear, 1 - Classified cloudy, 2 - Not classified, all other values undefined; not used in SIF Lite processing |
| co2_ratio | Ratio of CO2 retrieved in weak and strong CO2 band (value near 1 indicate scattering free scene) |
| delta_pressure_abp | Retrieved-predicted surface pressure from ABO2, usable as cloud screener; not used in SIF Lite processing |
| o2_ratio | Ratio of retrieved and predicted O2 column |
| surface_albedo_abp | Surface albedo (Lambertian equivalent) as retrieved in the ABO2 preprocessor at 760nm; not used in SIF processing |
| **Geolocation Group Variables** | |
| altitude | Surface altitude of observed footprint |
| footprint_latitude_vertices * | Latitude corner coordinates of the sounding location |
| footprint_longtitude_vertices * | Longitude corner coordinates of the sounding location |

| latitude | Center latitude of the measurement |
|---|---|
| longitude | Center longitude of the measurement |
| sensor_azimuth_angle | Azimuth angle between line of sight and local north |
| sensor_zenith_angle | Sensor zenith angle is the angle between the line of sight to the sensor and the local vertical |
| solar_azimuth_angle | Azimuth angle between the solar direction as defined by the sounding location, and the sounding local north |
| solar_zenith_angle | Solar zenith angle is the angle between the line of sight to the sun and the local vertical |
| time_tai93 | Timestamp (seconds since 1 January 1993) |
| **Metadata Group Variables** | |
| BuildID | The ID of the Build, including the software version that created this product |
| CollectionLabel | The Collection Label of the Build, including the software version that created this product |
| FootprintID * | OCO-2 footprint identifier (1-8), identifying the 8 independent OCO-2 spatial samples per frame |
| MeasurementMode | OCO-2/3: Instrument Measurement Mode, 0=Nadir, 1=Glint, 2=Target, 3=AreaMap, 4=Transition; users might consider separating these for analysis<br><br>GOSAT: Instrument Measurement Mode, 0=OB1D (FTS obs. mode I, sunlit), 1=OB2D (FTS obs mode II, sunlit), 2=SPOD (FTS specfic obs. mode, sunlit); users might consider separating these for analysis |
| OrbitID | Orbit Identifier: Start Orbit Number (OCO-2) or Start Solar Day (OCO-3) of observation<br><br>GOSAT: Orbit Identification String (\"NominalDay\|OrbitOfDay\|StartPathNumber-StopPathNumber\")" |
| SoundingID | Unique Identifier for each sounding |
| **Meteo (Meteorological) Group Variables** | |
| specific_humidity | Specific humidity at surface layer at the sounding location, interpolated from GEOS-5 FP-IT inst3_3d_asm_Nv field QV (specific_humidity); kg/kg |
| surface_pressure | Surface pressure at the sounding location; interpolated from GEOS-5 FP-IT inst3_3d_asm_Nv field PS (surface_pressure); Pa |
| temperature_skin | Skin temperature at the sounding location; interpolated from GEOS-5 FP-IT inst3_2d_asm_Nx field TS (surface_skin_temperature); K |
| temperature_two_meter | Two-meter temperature at the sounding location; interpolated from GEOS-5 FP-IT inst3_2d_asm_Nx field T2M (2-meter_air_temperature); K |
| vapor_pressure_deficit | Vapor pressure deficit at the sounding location (2m) (ECMWF forecast); Pa |
| wind_speed | Surface wind speed at sounding location; interpolated from GEOS-5 FP-IT inst3_2d_asm_Nx field U10M and inst3_2d_asm_Nx field V10M (10-meter_eastward_wind, 10-meter_northward_wind); m/s |
| **Offset Group Variables** | |
| SIF_Mean_757nm | Mean Solar Induced Fluorescence at 757nm (by footprint, for adjusted and unadjusted values) |





| | |
|---|---|
| SIF_Mean_771nm | Mean Solar Induced Fluorescence at 771nm (by footprint, for adjusted and unadjusted values) |
| SIF_Median_757nm | Median Solar Induced Fluorescence at 757nm (by footprint, for adjusted and unadjusted values) |
| SIF_Median_771nm | Median Solar Induced Fluorescence at 771nm (by footprint, for adjusted and unadjusted values) |
| SIF_Relative_Mean_757nm | Mean relative Solar Induced Fluorescence at 757nm (by footprint, for adjusted and unadjusted values) |
| SIF_Relative_Mean_771nm | Mean relative Solar Induced Fluorescence at 771nm (by footprint, for adjusted and unadjusted values) |
| SIF_Relative_Median_757nm | Median relative Solar Induced Fluorescence at 757nm (by footprint, for adjusted and unadjusted values) |
| SIF_Relative_Median_771nm | Median relative Solar Induced Fluorescence at 771nm (by footprint, for adjusted and unadjusted values) |
| SIF_Relative_SDev_757nm | Standard deviation of relative Solar Induced Fluorescence at 757nm (by footprint, for adjusted and unadjusted values) |
| SIF_Relative_SDev_771nm | Standard deviation of relative Solar Induced Fluorescence at 771nm (by footprint, for adjusted and unadjusted values) |
| signal_histogram_757nm | Signal level histogram for 757 nm radiances |
| signal_histogram_771nm | Signal level histogram for 771 nm radiances |
| signal_histogram_bins | Radiance level offset histogram bins |
| **Science Group Variables** | |
| continuum_radiance_757nm | Continuum Level Radiance at 757 nm |
| continuum_radiance_771nm | Continuum Level Radiance at 771 nm |
| daily_correction_factor | Correction factor to estimate daily average SIF from instantaneous SIF (using pure geometric incoming light scaling) |
| IGBP_index * | IGBP Index |
| SIF_757nm | Offset-Adjusted Solar Induced Chlorophyll Fluorescence at 757nm |
| SIF_771nm | Offset-Adjusted Solar Induced Chlorophyll Fluorescence at 771nm |
| SIF_Relative_757nm | Relative Solar Induced Fluorescence at 757 nm |
| SIF_Relative_771nm | Relative Solar Induced Fluorescence at 771 nm |
| SIF_Unadjusted_757nm | Solar Induced Chlorophyll Fluorescence at 757nm, no offset adjustment |
| SIF_Unadjusted_771nm | Solar Induced Chlorophyll Fluorescence at 771nm, no offset adjustment |
| SIF_Unadjusted_Relative_757nm | Solar Induced Chlorophyll Fluorescence at 757nm in fractions of continuum level, no offset adjustment |
| SIF_Unadjusted_Relative_771nm | Solar Induced Chlorophyll Fluorescence at 771nm in fractions of continuum level, no offset adjustment |
| SIF_Uncertainty_757nm | One-Sigma Statistical Uncertainty in Solar Induced Chlorophyll Fluorescence at 757nm |
| SIF_Uncertainty_771nm | One-Sigma Statistical Uncertainty in Solar Induced Chlorophyll Fluorescence at 771nm |





| sounding_land_fraction | Percentage of land surface type within the sounding |
| sounding_qual_flag | Sounding Quality Flag: 0 = good, 1 = bad |

500

**Table 2. Criterion of quality flags *best* and *good* for the Level 2 GOSAT, OCO-2, and OCO-3 data.**

Soundings that do not meet either set of criteria are flagged as *failed* (2).

| Quality_Flag = 0 (*best*) | Quality_Flag = 1 (*good*) |
|---|---|
| $28 \leq$ continuum radiance @757nm $\leq 195$ [W/m$^2$/sr/µm] | $28 \leq$ continuum radiance @757nm $\leq 195$ [W/m$^2$/sr/µm] |
| $\chi^2$ @ 757nm $\leq 2.0$ | $\chi^2$ @ 757nm $\leq 3.0$ |
| $\chi^2$ @ 771nm $\leq 2.0$ | $\chi^2$ @ 771nm $\leq 3.0$ |
| $0.85 \leq O_2$ ratio $\leq 1.5$ | $0.85 \leq O_2$ ratio $\leq 1.5$ |
| $0.5 \leq CO_2$ ratio $\leq 4.0$ | $0.5 \leq CO_2$ ratio $\leq 4.0$ |
| $\theta_{sun} \leq 80°$ for GOSAT; $\theta_{sun} \leq 70°$ for OCO-2/3 | $\theta_{sun} \leq 80°$ for GOSAT; $\theta_{sun} \leq 70°$ for OCO-2/3 |
| Land Fraction = 100% | Land Fraction $\geq 80\%$ |

503

504

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
