# Peer review of "Global GOSAT, OCO-2 and OCO-3 Solar Induced Chlorophyll"

_Earth System Science Data, 2021_

## Author Response (AR1)

We thank all reviewers for their time and effort to review our Data Description manuscript. We are very grateful for the time they have taken out of their busy schedule to help improve our manuscript.

**Referee #1:**

This manuscript presents a comprehensive description on the global L2 SIF Lite datasets of GOSAT, OCO-2 and OCO-3 satellites, which will be useful for users to fully utilize these data products. However, there are several issues that I still concern and recommend a major revision before publication.

Firstly, throughout the manuscript, the authors say that you describe the L2 SIF V10 data from GOSAT, OCO-2 and OCO-3. However, I don't find the GOSAT dataset following the Section 7, which gives the websites of these SIF datasets. Besides, only the version of OCO-2 data was v10. Therefore, what SIF dataset of GOSAT you used in this study? Please give detailed descriptions.

Response: Thank you for taking the time to carefully review our manuscript. Your comments and suggestions have greatly improved our data description.

We have revised the manuscript to clarify that the data described in our manuscript is V9 for GOSAT and v10 for OCO-2 and 3. We have also updated the data availability statement with a DOI for each of the three datasets. Thank you.

Secondly, as the authors said that the data description in this paper is an update and synthesis of several user guides, publications, and supplementary materials. Moreover, other works including bias correction, SIF retrieval methods, the calculation of SIF retrieval uncertainty and so on, were conducted like the same as the previous works. So what is the innovation of this work, or is there any new work compared to the previous work instead of a summary of the previous works?

Response: We want to emphasize that our manuscript was submitted as a Data Description. In short, we want to provide the data user community a formal, peer-reviewed publication that describes the GOSAT, OCO-2, and OCO-3 data itself and their differences and provides guidance on their use and interpretation. Such a document does not exist in any form, and is well within the ESSD scope for a Data Description manuscript.

Indeed, papers have been published on the retrieval method used to generate the data we describe, but no peer reviewed description of the Level 2 GOSAT, OCO-2, and OCO-3 data has been published. In fact, ESSD requests that the data descriptions not focus on the methods used to generate the data, but rather describe and present the data to promote the usability and accessibility of the data.

Our manuscript goes beyond the methods presented in previous manuscripts and guides by, as ESSD requires, 'highlighting and emphasizing the quality, usability, and accessibility of the dataset.' Our Discussion section, a sizable portion of the manuscript, is dedicated to exactly this requirement. The analyses, figures, descriptions, and discussions on how to properly use and interpret the data, given high sounding-level retrieval uncertainty and differences between sensors, are critically important to users who wish to use the data. Also, we offer harmonized quality control for improved cross sensor analysis, validation, and interpretation of global signals, that otherwise requires discrete ground samples.

We have expanded in the introduction upon the unique contribution that our manuscript provides:

"Our data description goes beyond previous documentation and publications via our description of the SIF Lite files and our presentation and comparison of the SIF data from the three platforms. Also, our discussions on SIF are intended to help the data user community to access and apply the data for scientific research and prevent misinterpretation."

Thirdly, the authors stated that the methods used to retrieve SIF were also outlined in the paper. However, the detailed SIF retrieval methods of three platforms were not presented in Section 4.1. Also, the differences of retrieval methods used on GOSAT and OCO-2/3 should also be given out.

Response: We revised Section 4.1 to clarify that the retrieval methods are identical for all platforms. We also moved the statement on the different retrieval windows in Section 3.1.2 to Section 4.1 SIF retrieval. The revised version of this section is below. Please also note that ESSD requests that data description manuscripts should not focus on methodology. Thus, we gave an overview of the SIF retrieval process because the specific details of the SIF retrievals have been previously peer reviewed and published.

"The SIF values provided in the SIF Lite files are based on spectral fits covering Fraunhofer lines, as SIF reduces the fractional depth of the Fraunhofer lines (Plascyk, 1975). The SIF retrieval methodologies are fully explained by Frankenberg et al. (2011b, a) and SIF is retrieved using the identical method for GOSAT and the OCO platforms at 757 nm and 771 nm. In brief, the main retrieval quantity in the retrieval state vector is the fractional contribution of SIF to the continuum level radiance, or relative fluorescence (SIF_Relative_757nm and SIF_Relative_771nm). The absolute SIF values (SIF_757nm and SIF_771nm) are generated during post-processing in W/m2/sr/μm.

It is important to note that although the SIF values have traditionally been loosely labeled as being retrieved at 757 nm and 771 nm, the retrieval fit windows used to produce the SIF Lite data is centered at 758.7 and 770.1 for OCO-2 and OCO-3, and at 758 and 771 for GOSAT.

However, we retain the 757 and 771 nomenclature to remain consistent with previous publications and to avoid confusion. We estimated SIF at 740 nm for each sounding using both retrieval windows as described in more detail below."

Finally, I think the conclusion is a little perfunctory. The conclusion seems to only emphasize the advantages of OCO-2/3 platforms. In addition, I don't find the innovative points in this manuscript.

Response: Thank you. We have added a leading sentence to the introduction to highlight the goal of our manuscript and the utility of the datasets we have provided:

"Here, we have presented and described the Level 2 SIF Lite files for GOSAT, OCO-2, and OCO-3, which have been standardized in the same netCDF format to maximize their interoperability and accessibility by the data user community and allow for intersensor comparisons."

Being that we have submitted a Data Description manuscript, we would be remiss to fail to describe the nature of the high-resolution data obtained by OCO-2 and OCO-3 and their observation modes as being innovative and advantageous. In fact, ESSD encourages authors to highlight and emphasize the quality of the data.

Specific:

Page 1:

Line 24-30: The authors may need to add some references here.

Response: Added a citation to Muller (1874).

Page2:

Line 32: How to judge whether it is extreme light condition? Please give detailed descriptions.

Response: To avoid confusion, we have removed this portion of the sentence.

Line 51-52: I can not understand the causal relationship between this sentence and the previous contents.

Response: We agree and have removed these two sentences.

Line 56-57: The authors should add the related references about GOSAT and OCO-2.

Response: Good catch, we have added the GOSAT citation (citations for OCO-2/3 were already present).

Page 2 Line 63-Page 4: I don't think these contents should belong to Introduction Section instead of Data description Section.

Response: Good suggestion, thank you. We have moved this content to the Data Description section.

Figure 1 and 2: We can observe obvious differences on sounding coverages between OCO-2 and OCO-3 from Figure 1 and 2. What differences on orbital imaging modes between them?

Response: We revised the figure description to clarify that the data shown is nadir mode.

Line 92-93: This description is not very suitable here as not only Fraunhofer lines but O2 absorption lines could also be used for SIF retrieval.

Response: We respectfully disagree with the reviewer on the usability of O2-absorption lines in the context of spaceborne SIF retrievals. In contrast to near-surface measurements, spaceborne SIF retrievals using O2 absorption bands are particularly challenging due atmospheric scattering (e.g., by clouds and aerosols) in the O2 A-band that can affect the TOA radiance similar to the SIF emission itself (Frankenberg et al., 2011). Also, there is an extinction (reabsorption) of the SIF emission through O2. In fact, all current satellite based SIF retrievals avoid O2 lines.

The sentence referenced by the reviewer is below and remains factually correct. However, we added "from space" to clarify that the application of O2 lines is not in general precluded.

"The retrieval of SIF from space requires high spectral resolution and a high signal to noise ratio (SNR) as solar Fraunhofer lines are very narrow and because SIF is a relatively weak signal (Frankenberg et al., 2011b)."

Line 97: Please check the unit of W/m2/sr/um throughout the manuscript.

Response: Thanks for catching the typos. They have been fixed.

Line 102-108: The introduction of spectral ranges of GOSAT was absent.

Response: We added this information:

"The sensor has four bands: 0.758-0.775 µm, 1.56-1.72 µm, 1.92-2.08 µm, and 5.56-14.3 µm."

Line 106, 113: Please point out the referenced radiance level and specified wavelength of the given SNRs.

Response: We have removed the reported SNR from the manuscript as specifics on how SNR is calculated is convoluted and gets far into the weeds for the purposes of our manuscript. It is also a bit unclear what the reviewer is requesting regarding the radiance level, and we did specify that the SNRs were for the oxygen A-band. We forwarded this comment from the reviewer to David Crisp, who authored the OCO manuscript we cited regarding the SNR. Here is his response:

It is a little bit convoluted. Here, I would focus on the A-band, because the focus of the paper is SIF. The > 400 value quoted in that paper is based on the l1b variable "snr_o2_l1b". This is actually an average of the SNR for continuum regions, defined as those regions falling between the 98th and 99th percentile for signal level in the band. (see the Data User Guide). I am not sure of what the reviewer is requesting for the reference radiance level. That is usually a term used for specifying instrument requirements. For example, OCO and OCO-2 were required to have a SNR > 290 at a signal level of 5.9 x 10^19 photons/sec/m^2/sr/micron. It exceeded that requirement. Another way to specify this is in terms of the "maximum measurable signal." Frankenberg et al. (AMT 2015) shows that a SNR of 400 corresponds to a ~10% of the maximum measurable signal, where the maximum measurable signal is 7 x 10^20 photons/sec/m^2/sr/micron. A SNR of 400 therefore relates to 7 x 10^19 photons/sec/m^2/sr/micron.

Line 113: signal-to-noise ratio ->SNR; please check the font format throughout the manuscript.

Response: Thanks, we have corrected SNR. The font seems to have changed during conversion to PDF, so we will double check this during the proofing stage. Thanks.

Page 8:

Line 205-209: Please provide the detailed criterion thresholds instead of criterion variables only.

Response: Thank you for the suggestion. We provide detailed criterion thresholds in Table 2. Each flag has 7 thresholds, which would be difficult to read and reference if this information were put into paragraph form. We have opted to keep this information in Table 2 for ease and to avoid redundancy.

Page 9:

Line 222-223: Actually, the spectral shape of SIF spectrum is not invariant and will vary with physiological factors and canopy structure, etc.

Response: This is a good point and we do recognize that the SIF spectrum can vary. However, please note that two previous publications by our coauthors have found SIF at wavelengths > 740 nm to be relatively invariant and to have a negligible impact on spaceborne SIF retrievals. This section now reads as:

"The spectral window in which SIF retrievals are made depends on the wavelength bands of the platform. Assuming the spectral shape of SIF is known and invariant, one can convert SIF to a standard reference wavelength. Here, we use 740 nm as a reference as it corresponds to the 2nd SIF peak and is not as strongly affected by chlorophyll re-absorption as red SIF, thus showing a relatively stable shape at wavelengths above 740 nm (Magney et al., 2019; Parazoo et al., 2019). The differences in the retrieval windows complicate the comparison of SIF retrievals from different sensors, thus it is useful to provide SIF at a well-defined reference wavelength.

Although the range of the wavelengths used to retrieve SIF from the various sensors is small (740-771 nm), absolute fluorescence can vary greatly depending on the spectral window used to retrieve SIF (Joiner et al., 2013; Köhler et al., 2018; Sun et al., 2018). However, reference far-red SIF emission spectra at the leaf level indicates that far-red fluorescence spectral shapes are consistent across species (Magney et al., 2019). Thus, we provide an estimate of absolute SIF740 (SIF_740nm) in the GOSAT and OCO-2/3 SIF Lite files derived from the empirical relationship between SIF at 740 nm and SIF at 758.7 nm and 770.1 nm (denoted as 757 nm and 771 nm; Eq. 1). The rationale for including SIF740 in the SIF Lite files is to allow for more consistent and robust comparisons of SIF and SIF-based analyses across sensors (Parazoo et al., 2019), and to reduce the retrieval error by a factor of 2 (Sun et al., 2018). We stress that the reported SIF740 values are not retrieved, but are estimated under the assumption that the spectral shape of SIF is invariant."

Eq.1: What is the basis for the chosen ratios? Please give a detailed analysis.

Response: We have clarified the basis for Eq. 1 by changing the sentence following Eq. 1 to:

"The ratios used in Eq. 1 were based on leaf level measurements conducted by Magney et al. (2019), however we observed a median ratio of 1.45 from OCO-2 over vegetated areas for 2015-2019 (Figure S1)."

Page 11:

Line 274: Why you chosen a 0.20-degree?

Response: We justified why we chose 0.20-degree spatial resolution as follows:

"...so that the barren surface reference targets had a coarser resolution than the soundings."

Line 291: ere?

Response: Thank you, this should be where

Line 289,294: The expressions of SIFdaily and Daily_SIF puzzled the readers.

Response: $SIF_{Daily}$ the term normally used to express daily average SIF when discussed in text or presented as a variable in an equation, as we did in Eq. 3. The Daily_SIF refers to the SIF Lite file variable name. We want data users to understand that the Daily_SIF variable in the nc file can be calculated as a product of SIF and the daily_correction_factor. We clarified that these terms are from the variables in the SIF Lite file.

"In terms of the SIF Lite file variable names, this equation can be written for SIF at any wavelength as Daily_SIF=SIF · daily_correction_factor."

Page 14:

Section 5.4: What are the differences between gridding the data and averaging the data soundings at large spatial scales? In my opinion, the sounding-level information could not be used after averaging them in large spatial and temporal scales.

Response: Great question! In a previous manuscript, we demonstrated how to use the Level 2 ungridded data and the benefits of retaining sounding-level information at large spatial scales. Using this manuscript as an example, we added this statement to clarify our meaning:

"For instance, as demonstrated by Doughty et al. (2019), ungridded Level 2 SIF data was used to calculate mean SIF for the entire Amazon Basin at different phase angles to show that the seasonality of SIF in the Amazon Basin was consistent across sun-sensor geometries. Such an analysis would not have been possible with gridded data because after gridding it is impossible to group the data by sounding-level attributes, such as phase angle or cloud fraction."

Page 15:

Line 378: "the use of SIF757 at would be more…"?

Response: Typo was fixed, thanks.

Figure 5: What temperature data were used? And what the basis for selection of temperature of 5°as a threshold? Please give out the detailed information.

Response: We used the temperature_skin variable in the SIF Lite data, which comes from the GOES-5 FP-IT 3h forecast, as we described twice in Section 3.1.3 and Table 1. We also stated that we chose the temperature threshold to filter for retrievals over vegetation where photosynthesis was likely to be occurring. We added the italicized text to the existing sentence that references Figure 5:

*"We progressively filtered the data as illustrated in Figure 5 to ensure the soundings were of a vegetated land surface, had similar sun-sensor geometries, environmental, and atmospheric conditions, and that the temperature was high enough for photosynthesis to occur as indicated by the temperature_skin variable in the SIF Lite data."*

Figure 5 and 7: Please modify the superscript format of R2 in the figures throughout the manuscript.

Response: These edits have been made, thanks.

Figure 8: Why you chosen the 4.0 degree?

Response: At finer spatial resolutions, we would have gridcells with no data because of the infrequent sampling of the GOSAT platform. We added this sentence to the main text description of Figure 8:

"We presented the comparison here at 4.0-degree spatial resolution to improve the sampling of GOSAT (Fig. 1a)."

**Referee #2:**

The authors described the global Level 2 SIF Lite data products for the Greenhouse Gases Observing Satellite (GOSAT), the Orbiting 14 Carbon Observatory-2 (OCO-2), and OCO-3 platforms. This study provides valuable SIF products, hence is of high interest for the SIF and remote sensing community. I recommend a major revision before the acceptance of the paper.

Response: Thank you for taking the time to carefully review our manuscript. Your comments and suggestions have greatly improved our data description.

General comments:

1. For introduction section: the authors described the global Level 2 SIF Lite data products from GOSAT, OCO-2 and OCO-3 platforms. However, the necessity and importance of harmonizing such a dataset was not adequately introduced in introduction section. Why these three sensors? Why not including GOME-2, TROPOMI? So, I suggest to restructure the introduction section to give clear explanations of the inventive of this study.

Response: The three datasets we described are official NASA data products and we are the team that performs the SIF retrievals, produces and quality checks the data, and updates the processing and retrieval algorithms (as alluded to by our affiliations). The GOME-2 and TROPOMI SIF data, which are produced from data gathered by satellites launched by the European Space Agency (ESA), have their own data processing pipelines and data file structures.

Here, we focus on high precision NASA products based on (1) high spectral resolution sensors, and (2) retrieval techniques using very small microwindows centered at 757 and 771 nm around the O2-A band. Comparison to broadband sensors can be found in Parazoo 2019 and is beyond the scope of this manuscript.

Respectfully, we want to reiterate that we submitted a Data Description manuscript, and thus our manuscript is not a 'study' per se. As stated in the abstract and elaborated in the introduction,

"Here, we describe, compare, and discuss the Level 2 SIF Lite version 10 (v10) data produced from three spaceborne platforms: the Greenhouse Gases Observing Satellite (GOSAT; http://dx.doi.org/10.22002/D1.8771), the Orbiting Carbon Observatory-2 (OCO-2), and OCO-3 (OCO-2 Science Team et al., 2020; OCO-3 Science Team et al., 2020). Our data description goes beyond previous documentation and publications via our description of the SIF Lite files and our presentation and comparison of the SIF data from the three platforms. Also, our discussions on SIF are intended to help the data user community to access and apply the data for scientific research and prevent misinterpretation."

2. For the introduction of Satellite platforms, I suggest to add a table to show the main specifications of the sensors.

Response: Thank you for the suggestion, and we did consider adding this information to a table. However, we decided to keep this information in paragraph form as each platform has different attributes that are important to discuss. For example, it is important to discuss how the mounting of OCO-3 to the ISS affects data acquisition relative to the other platforms. Putting the specifications discussed in sections 2.1 and 2.2 into a table would duplicate the information and thus force us to remove sections 2.1 and 2.2.

3. For Methods section: the detailed introduction of the SIF retrieval methods is missing. Several papers on the SIF retrievals for the platforms introduced in this paper have been published. Some of them are for simulated data, or improved compared with earlier versions of the SIF product. Which ones are adopted for the current products? So, it's better to give more details on the retrieval methods, as well as the quality control of the products in the paper.

Response: Reviewer 1 had the same concern, but the retrievals for GOSAT, OCO2, and OCO3 are identical. We revised Section 4.1 to clarify that the retrieval methods are identical for all platforms. We also moved the statement on the different retrieval windows in Section 3.1.2 to Section 4.1 SIF retrieval. The revised version of this section is below. Please also note that ESSD requests that data description manuscripts should not focus on methodology. Thus, we gave an overview of the SIF retrieval process because the specific details of the SIF retrievals have been previously peer reviewed and published.

As for quality control, we have documented the thresholds used to determine the quality of the data at the sounding level. Please see *Section 3.2 Quality flag criteria and rationale* and also Table 2.

"The SIF values provided in the SIF Lite files are based on spectral fits covering Fraunhofer lines, as SIF reduces the fractional depth of the Fraunhofer lines (Plascyk, 1975). The SIF retrieval methodologies are fully explained by Frankenberg et al. (2011b, a) and SIF is retrieved using the identical method for GOSAT and the OCO platforms at 757 nm and 771 nm. In brief, the main retrieval quantity in the retrieval state vector is the fractional contribution of SIF to the continuum level radiance, or relative fluorescence (SIF_Relative_757nm and SIF_Relative_771nm). The absolute SIF values (SIF_757nm and SIF_771nm) are generated during post-processing in W/m2/sr/μm.

It is important to note that although the SIF values have traditionally been loosely labeled as being retrieved at 757 nm and 771 nm, the retrieval fit windows used to produce the SIF Lite data is centered at 758.7 and 770.1 for OCO-2 and OCO-3, and at 758 and 771 for GOSAT. However, we retain the 757 and 771 nomenclature to remain consistent with previous publications and to avoid confusion. We estimated SIF at 740 nm for each sounding using both retrieval windows as described in more detail below."

SIF at 740 nm was estimated by SIF at 757 and 771 nm with Eq. (1) and included in the products. But how were the coefficients obtained? It's not clear. Even if there is a relation among them as shown in Eq. (1), the coefficients are expected to change with plant types, solor irridiance, environmental conditions etc. What's more, there is a lack of quatitative evaluation of the estimated SIF at 740 nm. A possible solution is to compare the estimated SIF@740 with the retrieved SIF@740 by TROPOMI, which can be used as a reference.

Response: Regarding the coefficients, Reviewer 1 had the same question. Here was our response:

We have clarified the basis for Eq. 1 by changing the sentence following Eq. 1 to:

"The ratios used in Eq. 1 were based on leaf level measurements conducted by Magney et al. (2019), however we observed a median ratio of 1.45 from OCO-2 over vegetated areas for 2015-2019 (Figure S1)."

Reviewer 1 also had the same comment regarding the invariance of the SIF line shape, and here was our response:

This is a good point and we do recognize that the SIF spectrum can vary. However, please note that two previous publications by our coauthors have found SIF at wavelengths > 740 nm to be relatively invariant and to have a negligible impact on spaceborne SIF retrievals. This section now reads as:

"The spectral window in which SIF retrievals are made depends on the wavelength bands of the platform. Assuming the spectral shape of SIF is known and invariant, one can convert SIF to a standard reference wavelength. Here, we use 740 nm as a reference as it corresponds to the 2nd SIF peak and is not as strongly affected by chlorophyll re-absorption as red SIF, thus showing a relatively stable shape at wavelengths above 740 nm (Magney et al., 2019; Parazoo et al., 2019). The differences in the retrieval windows complicate the comparison of SIF retrievals from different sensors, thus it is useful to provide SIF at a well-defined reference wavelength.

Although the range of the wavelengths used to retrieve SIF from the various sensors is small (740-771 nm), absolute fluorescence can vary greatly depending on the spectral window used to retrieve SIF (Joiner et al., 2013; Köhler et al., 2018; Sun et al., 2018). However, reference far-red SIF emission spectra at the leaf level indicates that far-red fluorescence spectral shapes are consistent across species (Magney et al., 2019). Thus, we provide an estimate of absolute SIF740 (SIF_740nm) in the GOSAT and OCO-2/3 SIF Lite files derived from the empirical relationship between SIF at 740 nm and SIF at 758.7 nm and 770.1 nm (denoted as 757 nm and 771 nm; Eq. 1). The rationale for including SIF740 in the SIF Lite files is to allow for more consistent and robust comparisons of SIF and SIF-based analyses across sensors (Parazoo et al., 2019), and to reduce the retrieval error by a factor of 2 (Sun et al., 2018). We stress that the reported SIF740 values are not retrieved, but are estimated under the assumption that the spectral shape of SIF is invariant."

Regarding the TROPOMI comparison, we must admit we do not see merit in such a comparison. First, the reviewer has assumed that SIF is retrieved at 740 nm for the TROPOMI data and could thus be used to validate SIF at 740 nm for GOSAT and OCO, but actually SIF is retrieved from TROPOMI using the 743 to 758 nm window and SIF at 740 nm in the TROPOMI data is estimated using the same method used for the GOSAT and OCO data (Kohler et al. 2018).

Second, Kohler et al. (2018) has already provided a comprehensive comparison of the TROPOMI and OCO-2 data, which is especially challenging given that to conduct an apples-to-apples inter-sensor comparison, the overpass time, day, sun-sensor geometry, spatial sampling, and cloud conditions must be as identical as possible - and there must be many soundings available for comparison because SIF is inherently noisy.

It should be noted that Philipp Kohler, who produces the TROPOMI SIF data from Caltech, is part of the team that produces the GOSAT and OCO SIF data and has contributed substantially to the writing and revision of our manuscript as a coauthor.

Specific comments:

line 11, second 'has' -> have;

Response: The subject of this sentence is the singular noun 'interest', thus we used the third-person singular form of the verb 'has'. For us to use the plural form 'have', the subject of our sentence must be plural. We have opted to not make the suggested change to maintain proper grammar.

line 49, PAM is not limited to measuring steady-state Fs;

Response: This statement was removed.

line 55, are all the three SIF products version 10?

Response: Actually, there will not be a version 10 of the GOSAT data. Thank you for bringing this to our attention.

This sentence now reads:

"Here, we describe, compare, and discuss the Level 2 SIF Lite version 9 (v9) data produced from the Greenhouse Gases Observing Satellite (GOSAT; http://dx.doi.org/10.22002/D1.8771), and Level 2 SIF Lite version 10 (v10) data from the Orbiting Carbon Observatory-2 (OCO-2), and OCO-3 (OCO-2 Science Team et al., 2020; OCO-3 Science Team et al., 2020)."

lines 66-67, what is the spatial resolution of the gridded data?

Response: Many users do not know the difference between the different levels of data, and often gravitate to using only gridded data, which is usually described as Level 3. The goal with this paragraph was to describe the differences in the levels for the user so that they have a clear understanding of what data our manuscript is describing.

There are multiple Level 3 SIF products, which have different spatial and temporal resolutions, and they are available at our website. The goal with this statement is to point users to the page in the event that they are interested in Level 3 (gridded) data. The goal of our manuscript is to describe Level 2 data (ungridded) and describing the various Level 3 products here would be out-of-context. Besides, the link features a table immediately at the top of the page that contains this information. Please take a look: https://climatesciences.jpl.nasa.gov/sif/download-data/level-3/.

lines 71-75, why the introduction of GOSAT is missing?

Response: This is not a statement that introduces the platforms. Here we describe who produces the Level 2 data for GOSAT and OCO-2/3, who does quality control, and where the data can be found. The GOSAT data is produced by the OCO team as there is no GOSAT-specific SIF team. We have clarified the beginning of the paragraph:

"The annual and monthly spatial distribution of the GOSAT and OCO Level 2 data for the globe and the continental United States are presented in Figures 1 and 2 for visualization. These data are produced by the OCO-2 and OCO-3 projects at the Jet Propulsion Laboratory (Frankenberg et al., 2014), quality controlled by NASA's Making Earth System Data Records for Use in Research Environments (MEaSUREs) SIF team, and are publicly available on the NASA Goddard Earth Sciences Data and Information Services Center (GES-DISC) website (https://disc.gsfc.nasa.gov/)."

line 81, why are some parts of the figure notes in bold? Please be consistent. same proble for all the figures;

Response: We followed the ESSD Word template, which has figure and table titles in bold. Thus, following the journal's template, we bolded these titles and were consistent in doing so. Please see https://www.earth-system-science-data.net/submission.html#templates for more details. We will defer the suggested changes to the ESSD editorial and proofing team.

line 89, it's not clear to me why Figures 1 & 2 are put here;

Response: All figures were placed after the paragraph in which they were referenced. We have moved all figures to appear after the main text.

line 92, signal to noise ratio -> signal-to-noise ratio;

Response: Changed, thanks.

line 93, remove 'because';

Response: Accepted, thanks.

line 97, measurement precision or retrieval error? how do you get this value (0.5 W/m2/sr/um)? 's' should be 'sr';

Response: Typo in the units were fixed. We clarified that this value is a mean:

"...enabling SIF retrievals with a mean single measurement precision around ~0.5 W/m2/sr/μm."

line 104, the references are cited twice;

Response: Fixed, thanks.

line 105, I suggest to also provide the spectral resolution in nm, or convert cm-1 to nm;

Response: We made the change, thanks.

line 199, s -> sr in the unit;

Response: Fixed.

line 203, criterion -> criteria;

Response: Good catch, thanks. Fixed.

line 211, section 4.1, please check my comments above;

Response: Please check our response above.

line 219, s -> sr in the unit;

Response: Fixed.

lines 233-234, how did you get the empirical relationship?

Response: We clarified these values in response to Reviewer 1:

"The ratios used in Eq. 1 were based on leaf level measurements conducted by Magney et al. (2019), however we observed a median ratio of 1.45 from OCO-2 over vegetated areas for 2015-2019 (Figure S1)."

lines 240-241, it's not clear to me how these numbers are used in Eq. 1;

Response: Please see our response above.

line 249, why 1-$\sigma$ represents the random component of the retrieval errors? according to Eq. 2 and lines 255-256, it represents the instrument noise;

Response: We apologize, the ", which" was misinterpreted as pertaining to the entire statement rather than the preceding variable. We have edited the description as:

"...and S0 is the measurement error covariance matrix and characterizes the instrument noise per detector pixel."

line 258, any sources for these numbers?

Response: Yes, the OCO data. We clarified:

"For the OCO-2/3 data, the uncertainty for SIF757 usually ranges between 0.3 and 0.5 W/m2/sr/μm, or ~15-50% of the absolute SIF value."

line 263, can you explain how you obtained Eq. 3?

Response: Sure, we have revised the description for this equation as:

"Uncertainty for $SIF_{740}$ is calculated from using the general formula for error propagation and the partial derivatives for the uncertainties for $SIF_{757}$ and $SIF_{771}$:"

line 265, section 4.4 Bias/offset correction, why did you carry out this correction? This correction was not performed for the original GOSAT and OCO-2/3 SIF products?

Response: As we described, biases in retrieved SIF can occur due to uncertainties in the exact instrument line-shape per footprint or slight uncertainties in detector linearity. The correction has always been applied to the SIF data.

lines 270-271, why do you choose these days for GOSAT and OCO-2/3?

Response: We added an explanation:

"These windows were chosen to obtain a robust background signal given their respective spatial-temporal resolution."

line 291, ere -> where;
Response: Fixed.

line 349, Figure 3, legends are illegible;

Response: This is due to PDF conversion and full resolution (600DPI) images will be provided during proofing.

lines 378-379, rephrase; how do you define 'weak'?

Response: Rephrased as, "It is important to note that in areas where the SIF signal is near zero…".

line 435, are the top panels for instantaneous SIF, and bottom panels for Daily SIF? these should be explained in the figure note;

Response: Great suggestion, thanks. The new title is:

"Figure 7. Relationships between instantaneous (top) and daily (bottom) SIF740, SIF757, and SIF771 from GOSAT and OCO-2 using Deming regression."

line 446, Figure 8, I would suggest add more numbers for the color scales, especially for the map of ratio.

Response: Great suggestion, thanks. We have added additional numbers to the color scales.

**Referee #3:**

The authors provided a detailed descriptions on three global SIF data products (GOSAT, OCO-2, OCO-3) in terms of SIF retrieval, quality flag, sun-sensor geometry and so on. Although the information was necessary for the correct use and interpretation of SIF data, many aspects have been presented or discussed in previous works and user guide. Moreover, it seems like that these three SIF products were independent besides they were retrieved in similar retrieval windows or similar spectral resolutions. Overall, the novelty of this manuscript is not clear, so I recommended a major revision before publication. The authors should provide some new findings rather than the simple summary of previous works.

Response: Thank you for taking the time to carefully review our manuscript. Your comments and suggestions have greatly improved our data description.

We want to emphasize that our manuscript was submitted as a Data Description. In short, we want to provide the data user community a formal, peer-reviewed publication that describes the GOSAT, OCO-2, and OCO-3 data itself and their differences and provides guidance on their use and interpretation. Such a document does not exist in any form, and is well within the ESSD scope for a Data Description manuscript.

Indeed, papers have been published on the retrieval method used to generate the data we describe, but no peer reviewed description of the Level 2 GOSAT, OCO-2, and OCO-3 data has been published. In fact, ESSD requests that the data descriptions not focus on the methods used to generate the data, but rather describe and present the data to promote the usability and accessibility of the data.

Our manuscript goes beyond the methods presented in previous manuscripts and guides by, as ESSD requires, 'highlighting and emphasizing the quality, usability, and accessibility of the dataset.' Our Discussion section, a sizable portion of the manuscript, is dedicated to exactly this requirement. The analyses, figures, descriptions, and discussions on how to properly use and interpret the data, given high sounding-level retrieval uncertainty and differences between sensors, are critically important to users who wish to use the data. Also, we offer harmonized quality control for improved cross sensor analysis, validation, and interpretation of global signals, that otherwise requires discrete ground samples.

We have expanded in the introduction upon the unique contribution that our manuscript provides:

"Our data description goes beyond previous documentation and publications via our description of the SIF Lite files and our presentation and comparison of the SIF data from the three platforms. Also, our discussions on SIF are intended to help the data user community to access and apply the data for scientific research and prevent misinterpretation."

Other comments:

The structure of this manuscript was not well organized and was a bit imbalanced. For example, Sections 2.1 (GOSAT) and 2.2 (OCO-2/3) belong to the satellite platform, but the Section 2.3 (observation modes) could be better presented in Section 3 (data description).

Response: We appreciate the reviewer's attention here, but the observation modes are characteristics of the physical platforms themselves. For instance, the pointing mirror assembly (PMA) allows for the acquisition of snapshot area mapping (SAM) mode data from OCO-3, as we described in Section 2.3. The descriptions of the observation modes in Section 2.3 are descriptions of the physical ability of the platforms to acquire data in different modes, not the data itself.

More pointedly, the data for the different observation modes are a consequence of the physical mode of data acquisition. Thus, the observation modes are described in *Section 2 Satellite platforms* rather than *Section 3 Data description*. It is important for the user community to understand differences in the physical characteristics of the platforms lead to the acquisition of data in different ways.

In addition, Section 4.1 (SIF retrieval) and 4.3 (SIF retrieval uncertainty) could be combined one section to reduce the number of subsections.

Response: We prefer to keep these sections separate because they are distinctly different discussions and to delineate that the equations shown relate to retrieval uncertainty. However, we have swapped Section 4.3 with 4.2 to improve the flow.

Discussion is confusing and it seems like suggestions on use of SIF. I was also confused why putting so many efforts on the comparison of OCO-2 and GOSAT SIF (four figures in this subsection and only eight figures in the whole manuscript).

Response: Thank you for bringing this point of confusion to our attention. Perhaps the reviewer's confusion stems from a misunderstanding of the type of manuscript that we submitted, which was a data description. We want to provide the data user community a formal, peer-reviewed publication that describes the GOSAT, OCO-2, and OCO-3 data itself and their differences and provides guidance on their use and interpretation.

Our discussion section builds upon the description of the data in Section 3 by discussing its usability, as requested by ESSD for this manuscript format. We have seven subsections for Section 5 which tackles the use and considerations that must be taken into account when

working with the SIF data, including the scaling of SIF to GPP, negative SIF values, sun-sensor geometry, averaging over space and time, the use of SIF 740 nm, a comparison of the GOSAT and OCO2 data, and the collocation of soundings. We were not able to compare OCO-3 with GOSAT or OCO-2 due to a lack of coincident soundings, as we mentioned in the opening paragraph of Section 5.6.

Line 43. PAM was mainly used in leaf level. It could be better to write "in vivo at the subcellular and using PAM at the leaf level"

Response: Changed, thanks.

Line 51-51. It is not clear to compare SIF and photosynthetic yields. Compare SIF with GPP?

Response: In response to another reviewer, this portion of the paragraph was removed.

Section 2.3. I found the observations of target mod was reduced in OCO-2 v10 compared to v8. Could you address this here?

Response: Nice catch, we have added the following sentence to Section 2.3:

"Target mode data for OCO-2 is absent from the v10 SIF Lite files, but will be included in the v11 update."

Line 366. Although it could mitigate the effects of sun-sensor geometry averaging sounding for a point of interest over the entire repeat cycle (16-days for TROPOMI), the seasonal variation of vegetation cannot be ignored in this long temporal period, which should be pointed out.

Response: We must admit that the reviewer's comment is a bit confusing, as it is not clear what the reviewer means by 'the seasonal variation of vegetation cannot be ignored…". The utility of the SIF data is to track changes in vegetation, including seasonality. Thus, it should never be the goal to remove vegetation-related seasonal effects from the SIF data. Perhaps the reviewer can elaborate if they think there is an important change that needs to be made.

Line 373-376. More relevant background information should be added in this short paragraph. Why the gridding is unnecessary? How to directly use L2 data? What information would be lost? It will confuse me and other readers.

Response: Reviewer #1 had the same questions, and we are happy to clarify. In a previous manuscript, we demonstrated how to use the Level 2 ungridded data and the benefits of retaining sounding-level information at large spatial scales. Using this manuscript as an example, we added this statement to clarify our meaning:

"For instance, as demonstrated by Doughty et al. (2019), ungridded Level 2 SIF data was used to calculate mean SIF for the entire Amazon Basin at different phase angles to demonstrate that the seasonality of SIF in the Amazon Basin was consistent across sun-sensor geometries. Such an analysis would not have been possible with gridded data because after gridding it is impossible to group the data by sounding-level attributes, such as phase angle or cloud fraction."

Line 378. at 757 nm?

Response: Thanks, we made the change.

Line 379. Stronger at this wavelength than that at 771nm?

Response: Thanks, we made the change.

---

## Author Response (AR2)

Dear Dr. Yuyu Zhou and reviewers,

We are extremely grateful for your feedback, expertise, and the time you have spent helping us refine our data description. Below are our revisions and responses in blue to a couple of comments that arose after we submitted our major revision.

On behalf of all coauthors, thank you so very much,

Russell Doughty

Report #2 from Anonymous referee #1:

Page 5:
Line 106: The authors stated that SIF retrievals with a mean single measurement precision around ~0.5 W/m2/sr/um. I want to know what is the basis for precision of 0.5 W/m2/sr/um. Up to now, there is no validation results that could demonstrate the spaceborne SIF retrieval precision.

We fully describe SIF retrieval errors in Section 4.2, which is titled *SIF retrieval uncertainty*. The equations and methods are fully documented and described in this section.

We have edited the statement at line 106 to read:

"...enabling SIF retrievals with single measurement precision around ~0.5 W/m2/s/μm (as fully described in Section 4.2)."

Line 114: Please verify the spectral resolution of GOSAT. 0.012 is different with that introduced by previous works.

We appreciate the attention to detail, but we reported the resolution as being, "... a spectral resolution of 0.2 cm$^{-1}$." Perhaps the reviewer was looking at a previous version of the manuscript, as a reviewer asked us to report the spectral resolutions in cm$^{-1}$.

This number comes from the Kuze et al. (2009) paper that we cited. Here is an excerpt from the abstract of that paper:

"TANSO-FTS is capable of detecting three narrow bands (0.76, 1.6, and 2.0 μm) and a wide band (5.5–14.3 μm) with 0.2 cm$^{-1}$ spectral resolution (interval)."